# Language Bottleneck Models: A Framework for Qualitative Cognitive Diagnosis

## Abstract

Accurately assessing student knowledge is central to education. Cognitive Diagnosis (CD) models estimate student proficiency at a fixed point in time, while Knowledge Tracing (KT) methods model evolving knowledge states to predict future performance. However, CD and probabilistic KT models represent knowledge states via quantitative estimates of knowledge concept mastery, limiting expressivity, while deep learning-based KT methods prioritize predictive accuracy at the cost of interpretability. We propose Language Bottleneck Models (LBMs), a general framework for producing textual knowledge state summaries that retain predictive power. LBMs use an encoder LLM to produce minimal textual descriptions of a student's knowledge state, which a decoder LLM then uses to reconstruct past responses and predict future performance. This natural-language bottleneck yields human-interpretable summaries that go beyond the quantitative outputs of CD models and capture nuances like misconceptions. Experiments show zero-shot LBMs rival state-of-the-art CD and KT accuracy on synthetic arithmetic benchmarks and real-world datasets (Eedi and XES3G5M). We also show the encoder can be finetuned with reinforcement learning, using prediction accuracy as reward, to improve summary quality. Beyond matching predictive performance, LBMs reveal qualitative insights into student understanding that quantitative approaches cannot capture, showing the value of flexible textual representations for educational assessment.

## 1 Introduction

**Knowledge state modeling** A fundamental objective in education is accurately assessing what a student knows, identifying misconceptions, and understanding how their knowledge evolves over time (Posner et al., 1982; Larkin, 2012; Chen et al., 2020). Teachers intuitively achieve this through diagnostic reasoning: by observing students' answers, they infer not merely correctness, but deeper patterns reflecting conceptual mastery or specific misunderstandings.

**Limitations of Cognitive Diagnosis and Knowledge Tracing** Cognitive Diagnosis (CD) (Templin et al., 2010; Wang et al., 2024) models produce diagnostic reports but are constrained to proficiency estimates over a fixed set of knowledge concepts. In parallel, Knowledge Tracing (KT) (Shen et al., 2024) models excel at predicting future performance based on observed past responses, yet remain limited to either estimating mastery over knowledge concepts (Corbett & Anderson, 1994; Zhou et al., 2024) or operating as black-box models requiring post-hoc interpretability (e.g. DKT (Piech et al., 2015; Ghosh et al., 2020)).

**Limitations of existing LLM-based approaches** Recent approaches leveraging large language models (LLMs) have shown that LLMs can produce sensible predictions of students' future behavior when provided with relevant information (Li et al., 2024a; Kim et al., 2024), help mitigate the KT's cold-start problem (Lee et al., 2024) and be finetuned to improve accuracy on KT tasks (Wang et al., 2025). Nonetheless, these LLM-based methods still lack rigorous interpretability, as they either treat the model as a black-box or rely on free-form explanations that are susceptible to hallucination (Bender et al., 2021). As a result, such methods fail to provide grounded, reliable representations of knowledge that can be trusted in educational practice. This gap motivates a principled approach where interpretability is an intrinsic design constraint.

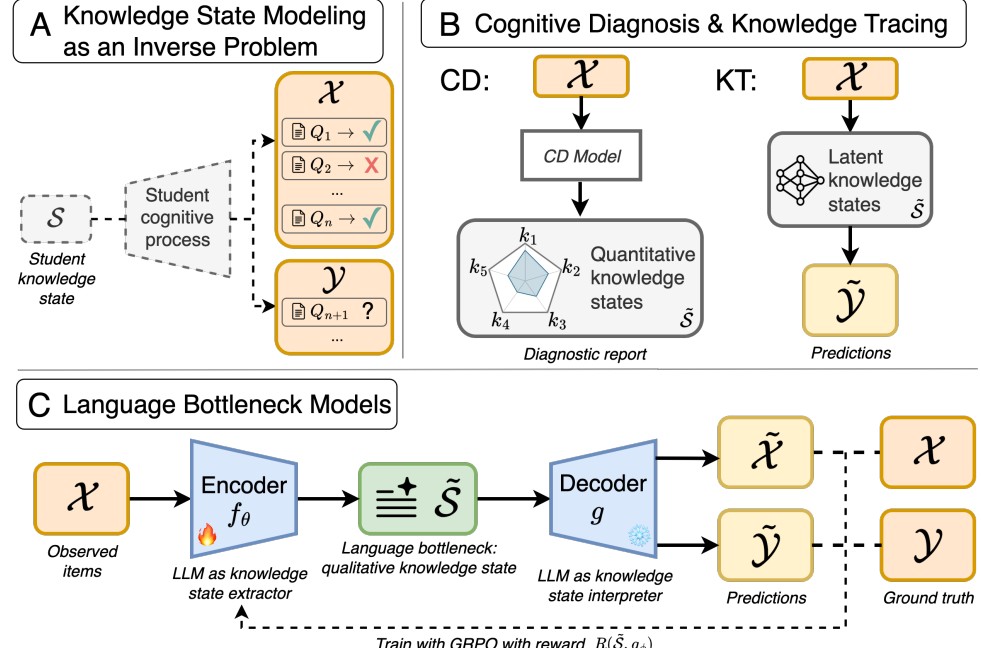

Figure 1: **Language Bottleneck Models for Knowledge Modeling.** (A) Past and future behavior $\mathcal{X}$ and $\mathcal{Y}$ are caused by a certain knowledge state $\mathcal{S}$ held by the student when answering questions. (B) CD and KT models represent the knowledge state via quantitative proficiency vectors or opaque latent embeddings. (C) LBMs approximate the knowledge state using natural language summaries which are used to predict past and future behavior.

**Knowledge state modeling as an inverse problem**   Assessing student knowledge can be framed as an *inverse problem* (Figure 1A): observable answers are generated by an underlying knowledge state $\mathcal{S}$, and the goal is to infer a faithful approximation $\tilde{\mathcal{S}}$. CD models constrain $\tilde{\mathcal{S}}$ to fixed quantitative proficiency estimates over knowledge concepts (Figure 1B), providing interpretability but lacking flexibility to capture nuanced knowledge states. KT models prioritize predictive performance but have uninterpretable latent representations (Figure 1B). We adopt the inverse-problem framing but allow $\tilde{\mathcal{S}}$ to be expressed as concise, free-form natural language, so knowledge states remain predictive while avoiding the constraints of a fixed set of knowledge concepts.

**Language Bottleneck Models**   We introduce *Language Bottleneck Models (LBMs)* as a general framework for compressing a student's interaction history into a predictive, text-based knowledge state. As shown in Figure 1C, an encoder LLM maps a student's interaction history $\mathcal{X}$ into a natural language summary $\tilde{\mathcal{S}}$, while a frozen decoder LLM must reconstruct past responses and predict future ones using only that summary. In an educational setting, they constitute a language-based approach to Cognitive Diagnosis, not bound to fixed skill vocabularies and can flexibly describe nuanced insights such as misconceptions.

**Contributions.**

- We cast knowledge state modeling as an inverse problem over open-ended textual representations—building on ideas from Cognitive Diagnosis but replacing rigid concept proficiencies with a flexible natural-language knowledge state.
- To instantiate this, we introduce Language Bottleneck Models (LBMs), which encode observed student behavior into predictive text-based summaries of knowledge states.
- We extensively evaluate LBMs on synthetic and real-world datasets against 14 KT and CD baselines across 7 open- and closed-source LLM backbones, present a detailed case study of qualitative differences with CD knowledge states, and demonstrate that LBMs can be effectively trained and steered via their textual summaries.

## 2 KNOWLEDGE STATE MODELING AS AN INVERSE PROBLEM

### 2.1 PRELIMINARIES AND NOTATION

We consider data consisting of student interactions with educational questions. At each time step $t$, a student is presented with a question $q_t \in \mathcal{Q}$ and optionally knowledge concept (KC) information $k_t \in \mathcal{K}$ and provides a response $r_t \in \mathcal{R}$, which is evaluated for correctness $c_t \in \{0, 1\}$. We represent an interaction as $x_t = (q_t, k_t, r_t, c_t)$, and a student's interaction history up to time $t$ as $H_t = (x_1, \ldots, x_t)$.

The standard predictive task, often referred to as *Knowledge Tracing (KT)* (Corbett & Anderson, 1994; Shen et al., 2024), is to estimate $p(c_{t+1} \mid q_{t+1}, H_t)$, the probability that the student will answer a new question $q_{t+1}$ correctly, conditioned on their interaction history. This task definition captures the forward-prediction aspect of knowledge state modeling, but does not yet address how the underlying knowledge that drives these responses should be represented.

### 2.2 KNOWLEDGE STATE MODELING AS AN INVERSE PROBLEM

An alternative view is to frame knowledge state modeling as an *inverse problem*: observed responses are generated by a latent knowledge state $\mathcal{S}$ through the student's cognitive process, and the goal is to recover an approximation $\tilde{\mathcal{S}}$ of this state from the observed responses.

This perspective is central to *Cognitive Diagnosis (CD)* models (Reckase, 2006; De La Torre, 2009; Templin et al., 2010; Wang et al., 2022; 2024), which produce diagnostic reports of concept mastery from observed responses. However, CD typically restricts $\tilde{\mathcal{S}}$ to quantitative mastery or proficiency vectors based on predefined or inferred concepts, limiting expressivity. Meanwhile, deep learning-based KT methods prioritize predictive accuracy without explicitly recovering $\tilde{\mathcal{S}}$, representing knowledge states as high-dimensional embeddings that lack transparency (Piech et al., 2015; Zhang et al., 2017; Pandey & Karypis, 2019; Ghosh et al., 2020). Bayesian KT (Corbett & Anderson, 1994) as well as recent KT extensions introduce diagnostic reports similar in spirit to CD, or use interpretable latent states (Yeung, 2019; Minn et al., 2022; Chen et al., 2023; Park et al., 2024; Zhou et al., 2024). However these approaches remain bound to estimates of KC mastery or rely on post-hoc interpretability.

**Key assumption: constant knowledge state** A practical assumption underlying this formulation is that a knowledge state can be treated as constant within short diagnostic windows (e.g., unit tests, placement exams, or tutoring sessions). This aligns with CD models (see §2.1 in Wang et al. (2024)), and distinguishes them from KT approaches which model the evolution of $\mathcal{S}_t$ across longer periods where the underlying knowledge state is expected to change.

### 2.3 NATURAL LANGUAGE AS THE INTERFACE

Our formulation follows CD approaches in adopting the inverse problem framing of inferring a diagnostic report from observed responses, but instead of restricting $\tilde{\mathcal{S}}$ to quantitative mastery scores, we model it through concise textual summaries. Natural language provides an interpretable and expressive medium—capable of describing arbitrary reasoning patterns or misconceptions, a key focus in education research (Smith III et al., 1994; Wang et al., 2020; King et al., 2024)—while remaining human-understandable. In the next section, we introduce *Language Bottleneck Models (LBMs)*, which operationalizes this idea by compressing student interaction histories into concise textual representations that preserve predictive information.

## 3 LANGUAGE BOTTLENECK MODELS

### 3.1 FORMAL DEFINITION

We propose *Language Bottleneck Models* (LBMs) for Knowledge State Modeling via textual summaries: an LLM-based, two-stage architecture designed to infer a predictive text-based knowledge state from a student's interaction history.

Let $X^{\text{enc}} \subseteq H_t$ denote a subset of observed interactions used by the *encoder*. An encoder LLM $f_\theta$ maps this history to a natural-language summary: $\tilde{\mathcal{S}} = f_\theta(X^{\text{enc}})$. This summary serves as the sole representation of the student's knowledge state. A *decoder* LLM $g_\phi$ then conditions only on $\tilde{\mathcal{S}}$ to predict the probability that the student will answer a question $q \in \mathcal{Q}$ correctly: $g_\phi(q, \tilde{\mathcal{S}}) = p(c \mid q, \tilde{\mathcal{S}})$, that is, given a question $q$ and a summary $\tilde{\mathcal{S}}$, the decoder predicts the probability that the student will answer correctly.

In principle, both encoder $f_\theta$ and decoder $g_\phi$ could be trained. However, we show with the following motivating experiments that decoding is not the hard part: when given high-quality knowledge state summaries, off-the-shelf LLMs can achieve near-perfect prediction accuracy. For this reason, we limit the scope of this work to learning an encoder that produces faithful, predictive, and interpretable summaries, keeping $g_\phi$ fixed.

### 3.2 MOTIVATING OBSERVATIONS

We motivate the design of LBMs by two observations.

**Observation 1: Given a good knowledge state summary, strong LLMs can decode with high fidelity.** To test this in an idealized setting, we used a Synthetic dataset where each student's knowledge state is programmatically generated. Figure 2 evaluates the performance of different decoder models when given direct access to this perfect, "ground-truth" summary of the student's latent knowledge (example knowledge state and summaries are shown in Figure A1 in the Appendix, and full dataset details are in §5). Stronger models like GPT-4o achieve nearly perfect accuracy ($98\%$), indicating that the bottleneck representation is indeed sufficient to drive effective downstream prediction for closed-form questions —provided it captures the right information.

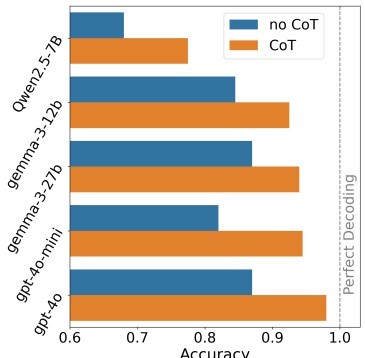

**Observation 2: Summarizing knowledge states from raw interactions is non-trivial.** Standard LLM summarization approaches can capture high-level skill mastery but often fail to identify crucial latent patterns like student misconceptions (see an example comparison on our synthetic dataset Figure A1 in the Appendix).

Figure 2: **Accuracy on synthetic dataset given ground truth knowledge state summaries.**

Together, these observations suggest that the key challenge lies in learning an encoder that produces faithful summaries, rather than in the decoding step itself.

### 3.3 TRAINING OBJECTIVE AND OPTIMIZATION

We propose a reinforcement learning-based approach to train an encoder to produce more faithful and predictive summaries by using downstream decoder accuracy as reward.

**Summary generation and prediction**   Given an interaction history $H_t = (x_1, \ldots, x_t)$, the encoder $f_\theta$ maps a subset of interactions $\mathcal{X}_{\text{enc}} \subseteq H_t$ to a textual summary $\tilde{\mathcal{S}} = f_\theta(\mathcal{X}_{\text{enc}})$. The frozen decoder $g$ then conditions on $\tilde{\mathcal{S}}$ to predict responses for two sets $\mathcal{X}$ and $\mathcal{Y}$: interactions used for reconstruction and prediction, respectively. In practice, $\mathcal{X}$ and $\mathcal{Y}$ may be chosen flexibly to include held-out past responses, future responses, or both.

**Reward function**   Given predicted interactions $\tilde{\mathcal{X}} = \{g(q, \tilde{\mathcal{S}}), q \in \mathcal{X}\}$ and $\tilde{\mathcal{Y}} = \{g(q, \tilde{\mathcal{S}}), q \in \mathcal{Y}\}$, the reward for a summary $\tilde{\mathcal{S}}$ is defined as

$$R(\tilde{\mathcal{S}}; g) = \phi\Big(\texttt{Acc}\big(\tilde{\mathcal{X}}, \mathcal{X}\big), \texttt{Acc}\big(\tilde{\mathcal{Y}}, \mathcal{Y}\big), |\tilde{\mathcal{S}}|, \Omega(\tilde{\mathcal{S}})\Big), \tag{1}$$

where $\texttt{Acc}(\cdot, \cdot)$ measures accuracy, $|\tilde{\mathcal{S}}|$ penalizes overly long summaries, and $\Omega(\tilde{\mathcal{S}})$ enforces optional structural constraints (e.g., inclusion of a $\texttt{Misconceptions}$ section). The function $\Phi$ balances these components using hyperparameters or indicator functions to enforce constraints as needed.

**Optimization via GRPO** We optimize the encoder $f_\theta$ with Group Relative Policy Optimization (GRPO) (Shao et al., 2024). For each input $\mathcal{X}_{\text{enc}}$, the encoder generates $G$ candidate summaries $\{\tilde{\mathcal{S}}^1, \ldots, \tilde{\mathcal{S}}^G\}$, each evaluated by $R(\tilde{\mathcal{S}}^i; g)$. We then compute group-relative advantage and update parameters:

$$A(\tilde{\mathcal{S}}^i) = \frac{R(\tilde{\mathcal{S}}^i; g) - \mu}{\sigma}, \quad \nabla_\theta J(\theta) = \frac{1}{G} \sum_{i=1}^{G} A(\tilde{\mathcal{S}}^i) \nabla_\theta \log p_\theta(\tilde{\mathcal{S}}^i \mid \mathcal{X}_{\text{enc}}), \tag{2}$$

where $\mu$ and $\sigma$ are the mean and standard deviation of rewards within the group.

### 3.4 STEERABILITY OF THE ESTIMATED KNOWLEDGE STATE

The natural-language summaries generated by LBMs allow for various human-model interactions (detailed in Appendix D). **(1) Prompt engineering the encoder.** Since the encoder $f_\theta$ is itself an LLM, its behavior can be shaped through prompt design, such as system instructions or in-context examples (Brown et al., 2020). **(2) Steering via reward signals.** Rewards to steer the encoder towards human preferences can be incorporated through the $\Omega(S)$ term in Eq. 1. **(3) Augmenting with student-specific information.** Educators can supplement the model with additional knowledge not present in observed data—either by augmenting encoder inputs or by editing the generated summary before decoding. This enables integration of recent classroom observations or specific misconceptions identified through in-person interactions.

## 4 RELATED WORK

We review related works from the Cognitive Diagnosis and Knowledge Tracing literature, as well as concept bottleneck models. See Appendix F for an extended review of related works, and Table F1 for a high-level comparison of LBMs with CD and KT.

**Cognitive Diagnosis** Cognitive Diagnosis Models (CDMs) infer student knowledge states from observed responses. Classical approaches include Item Response Theory (IRT) and Multidimensional IRT which measure continuous proficiency scores (Rasch, 1993; Reckase, 2006), and the DINA model and its variants which estimate binary mastery of knowledge concepts (De La Torre, 2009). Recent deep learning variants like NeuralCDM (Wang et al., 2022) and RCD (Gao et al., 2021) use neural networks and graph architectures to model complex relationships between students, questions, and knowledge concepts. However, these models typically operate within predefined or inferred knowledge frameworks and provide only quantitative skill mastery estimates.

**Knowledge Tracing** Knowledge Tracing methods model student learning to predict future performance. Deep learning approaches like DKT (Piech et al., 2015), DKVMN (Zhang et al., 2017) and AKT (Ghosh et al., 2020) employ neural architectures. Despite strong predictive performance, these models represent knowledge as abstract latent vectors, limiting interpretability. Several recent works have proposed more interpretable KT. Early Bayesian approaches (Corbett & Anderson, 1994; Käser et al., 2017) and recent probabilistic models (Minn et al., 2022; Zhou et al., 2024) learn interpretable latent states representing student proficiency over knowledge concepts, while IRT-based methods (Yeung, 2019; Chen et al., 2023) combine deep learning with item response theory for meaningful latent representations. Other works improve interpretability through learned question relationships (Tong et al., 2022), explainable subsequences (Li et al., 2023), or option tracing (Ghosh et al., 2021). However, these approaches remain fundamentally constrained to quantitative concept proficiency estimation or require post-hoc interpretability. Finally, recent LLM-based approaches have shown promise for knowledge tracing tasks (Li et al., 2024a; Wang et al., 2025), but they generally remain opaque, either treating LLMs as black boxes with no interpretable intermediate representation or relying on model-generated explanations susceptible to hallucination.

**Concept Bottleneck Models** Concept Bottleneck Models (CBMs) (Koh et al., 2020) improve interpretability by using human-understandable concept activations as intermediates between inputs and predictions. However, CBMs typically rely on finite predefined concept sets, limiting applicability to complex tasks like knowledge tracing. Recently, Explanation Bottleneck Models (XBMs) (Yamaguchi & Nishida, 2024) use textual rationales as intermediates for vision classification. While

our Language Bottleneck Models (LBMs) adopt this language bottleneck concept, they differ fundamentally: unlike XBMs' instance-specific rationales, LBM summaries capture implicit knowledge states that generalize to future, unknown questions, requiring holistic, adaptable summaries rather than task-specific rationales.

## 5 EXPERIMENTS

**Datasets** We evaluate LBMs and baseline models on a synthetic arithmetic benchmark and two real-world datasets (Table 1). Our Synthetic dataset (Appendix B.3.1) simulates learners answering addition, subtraction, multiplication, and division questions; each student is assigned mastered skills, unmastered skills, and systematic misconceptions. We filter *Eedi* and *XES3G5M* for single-session trajectories (max inter-question gaps of 3 and 10 minutes respectively) with at least 40 and 34 interactions respectively to approximate the static knowledge state assumption common in CD (Wang et al., 2024). We evaluate *XES3G5M* in both Chinese and English using translations from Ozyurt et al. (2024) (see Appendix B.3).

Table 1: Overview of datasets. AVG#log and STD#log>1 are defined following Wang et al. (2022) as respectively the average number of logs per student per KC, and the mean standard deviation of score per student and per KC.

| Dataset | Synthetic | Eedi (Filt.) | XES3G5M (Filt.) |
|---|---|---|---|
| #Students | 2,000 | 623 | 996 |
| #Questions | 5,000 | 3,401 | 3,221 |
| #KCs | 4 | 1,284 | 803 |
| #Logs | 420,000 | 28,947 | 57,788 |
| #Logs/stud. | 210 | $\geqslant$40 | $\geqslant$34 |
| AVG Acc. | 0.55$\pm$0.20 | 0.68$\pm$0.18 | 0.85$\pm$0.36 |
| AVG#log | 52.5$\pm$0.0 | 1.65$\pm$0.52 | 2.02$\pm$0.55 |
| STD#log>1 | 0.29$\pm$0.12 | 0.38$\pm$0.08 | 0.24$\pm$0.14 |

**Models** We evaluate LBMs across LLM backbones of different sizes and capabilities, both open-source (Qwen 2.5 3B and 7B (Team, 2024), Gemma 3 12B and 30B (Team, 2025)), and closed-source (GPT-4o-mini, GPT-4o and GPT-5 (Achiam et al., 2023)). Unless noted otherwise, we run the *instruct* variants of each open-source model, use the same backbone LLM for both the encoder and decoder, and prompt all models to provide their response directly without chain-of-thought. We run GPT-5 with `reasoning_effort=minimal` configuration. Hyper-parameters and prompt templates are provided in Appendix B.

**Baselines** We compare LBMs against 9 Knowledge Tracing methods: DKT (Piech et al., 2015), DKVMN (Zhang et al., 2017), SAKT (Pandey & Karypis, 2019), AKT (Ghosh et al., 2020), Deep IRT (Yeung, 2019), SAINT (Choi et al., 2020), SimpleKT (Liu et al., 2023a), QIKT (Chen et al., 2023), GKT (Nakagawa et al., 2019)) implemented with the PYKT library (Liu et al., 2022) and 5 Cognitive Diagnosis methods (IRT (Rasch, 1993), MIRT (Reckase, 2006), DINA (Junker & Sijtsma, 2001), KaNCD and NeuralCDM (Wang et al., 2022)) implemented with the EduCDM library (bigdata ustc, 2021). We also run each LLM via *direct prompting*, where the LLM predicts answers from the full interaction history without a bottleneck. Training details for all baselines are given in Appendix B.

**Systematic evaluation of knowledge state summaries** We systematically evaluate the quality of the knowledge state summaries produced by different encoder models on the Synthetic dataset using LLM-as-a-judge (Li et al., 2025). Specifically we prompt a GPT-5 model to compare each summary to the corresponding ground-truth knowledge state across the following dimensions: **Global score** (*overall alignment with the true state*); **Construct accuracy** (*correctly assessed constructs*); **Misconception detection** (*correctly identified student misconceptions*); **Misconception false positives** (*misconceptions mentioned but absent in the ground truth*); **Specificity** (*precision vs. vagueness of the description*); and **Confidence calibration** (*degree of over- or under-confidence*). See Appendix C for full details and prompts.

### 5.1 QUALITATIVE INSIGHTS

#### 5.1.1 CASE-STUDY: COMPARING CD AND LBM KNOWLEDGE STATE REPRESENTATIONS

A key advantage of LBMs over CD models is the ability to capture nuanced insights about the student knowledge state, such as misconceptions. We illustrate this with a case-study Figure 3. We train a state-of-the-art CD model (NeuralCDM) trained on our Synthetic dataset, which achieves strong predictive performance (AUC: 0.96, Acc: 0.90 on the test set). From this model, we extract proficiency estimates across knowledge concepts using the learned student embedding vector, producing a typical

CD diagnostic report. We then compare this to the knowledge state summary generated for the same student by an LBM (trained Gemma-12B encoder with frozen Gemma-27B decoder, as in §5.2).

### True Knowledge State

**Student #1474**   $\mathcal{S}$

**Mastered:**
Addition, subtraction, multiplication.
**Fails on:**
Division.
**Misconceptions:**
Fails operations involving 6 as operand,
Forgets to carry in addition,
Fails with negative numbers.

### Estimated Knowledge State

**NeuralCDM**   $\tilde{\mathcal{S}}$

| KC | Proficiency |
|---|---|
| Addition | 0.59 |
| Subtraction | 0.53 |
| Multiplication | 0.76 |
| Division | 0.23 |

**LBM**   $\tilde{\mathcal{S}}$

The student excels at addition and multiplication with integers [...]. Subtraction is a weakness, often producing incorrect answers, especially with negative results. Division [...] is consistently incorrect. Multiplication by 6 or 7 seems to be a specific area of difficulty, occasionally missed despite otherwise demonstrating mastery of multiplication.

Figure 3: **Case study: comparing CD and LBM knowledge states.** Given a student from the Synthetic dataset, we compare proficiency estimates across knowledge concepts (KCs) obtained from a trained NeuralCDM model to the text-based knowledge state generated by a trained LBM model.

While NeuralCDM reliably captures general KC proficiency, its estimates are influenced by misconceptions without explicitly identifying them. In contrast, LBMs capture overall proficiency and uncover specific misconceptions (e.g., errors with negative numbers or with operand-6). This ability to provide nuanced, qualitative insights into student knowledge states sets LBMs apart from CD methods.

#### 5.1.2 SYSTEMATIC EVALUATION OF KNOWLEDGE SUMMARIES

To further assess the interpretability of the knowledge states produced by LBMs, we systematically compare their summaries to ground truth knowledge states from the Synthetic dataset using GPT-5 as LLM-as-a-judge (Li et al., 2025). The results are shown Figure 4. Figure 4(a) shows the overall alignment with the ground truth summary (score from 1='mostly incorrect' to 5='strongly aligned'), and the construct mastery accuracy. Model capability correlates with performance on both metrics, with GPT-5 achieving the highest scores. Figure 4(b) plots the misconception-detection rate against the misconception false positive rate. Qwen2.5 and GPT-4o/4o-mini exhibit lower hallucination but also lower detection; Gemma-3 detects more misconceptions but at the cost of more false positives; GPT-5 outperforms all models on both dimensions. Figure 4(c) evaluates summary specificity and confidence calibration. Gemma-3 models produce more specific summaries than Qwen2.5, but both families tend to be over-confident relative to GPT models, with GPT-5 displaying the best overall specificity and calibration.

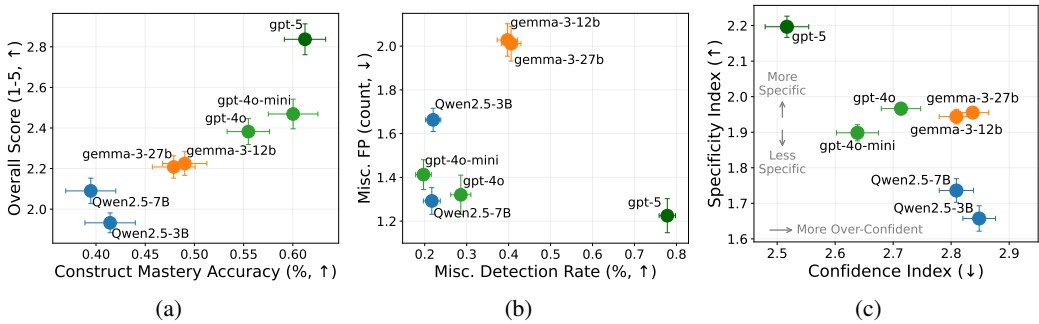

Figure 4: Systematic evaluation of the summaries produced by different encoder LLMs over 200 test students from the Synthetic dataset: (a) Average construct mastery accuracy across constructs (%) and average overall score (1-5 scale); (b) Average misconception detection rate (%) and average number of misconception false positives per summary (count); (c) Confidence Index (1-3, 1=under-confident, 2=appropriately calibrated, 3=over-confident) and Specificity Index (1-3, higher is more specific). Error bars represent the standard error (N=200).

## 5.2 TRAINING LBM ENCODERS

As outlined in §3.3, downstream accuracy can serve as a reward signal to train encoders to produce increasingly accurate summaries. We demonstrate this by training a Gemma3-12B encoder with GRPO alongside a frozen Gemma3-27B decoder on 800 students. We set the reward as the decoder accuracy across $|\mathcal{Y}| = 20$ unseen questions $R(\tilde{\mathcal{S}}; g) = \text{Acc}(\tilde{\mathcal{Y}}, \mathcal{Y})$, train with a LoRA adapter (Hu et al., 2022) and evaluate on 200 unseen test students.

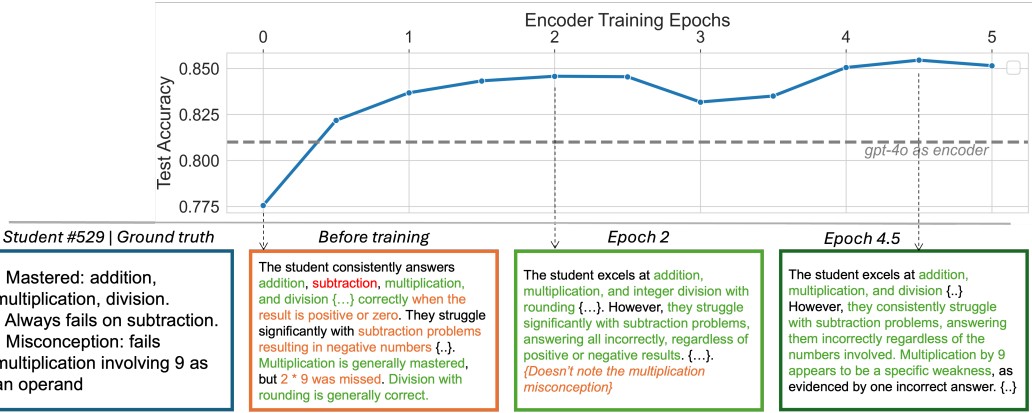

Figure 5: **Training the encoder with the decoding accuracy on the synthetic dataset.** Evolution of the test accuracy as a Gemma3-12B encoder is trained as described in § 3.3, with a fixed Gemma3-27B decoder. Trained on 800 training students and tested on 200 students, with $|\mathcal{X}| = 50$ questions per input trajectory and $|\mathcal{Y}| = 20$ questions to predict per student. The bottom row shows the evolution of the generated summary over the course of training for an example student. Text is colored green (exact), orange (approximate), or red (false) based on ground-truth.

**Effect on accuracy** Figure 5 shows the encoder progressively improving summary quality and quickly outperforming GPT-4o. The figure illustrates this through an example student who mastered all constructs except subtraction and fails any multiplication involving 9. The initial summary contains inaccuracies and misses this systematic misconception, while the final summary successfully captures the student's complete knowledge state. Stratifying by knowledge state complexity, we observe larger gains for more complex cases (Appendix A.3.1).

**Qualitative evolution of summaries** We use the same LLM-as-a-judge approach as in the previous section to systematically evaluate the generated summaries over training. Figure 6 shows that the summaries gradually better capture concept mastery and align more with the ground truth knowledge state over training. Moreover, as summaries also become more specific and better calibrated. Misconception detection and false positive rates do no significantly change over training (Appendix A.3.2).

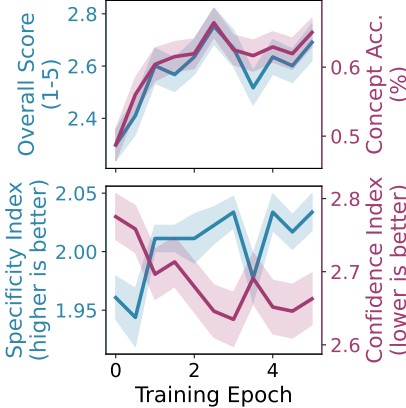

Figure 6: **Systematic evaluation of generated summaries over encoder training.** Evaluation axes are the same as Figure 4(a) and (c). Confidence intervals represent the standard error (N=200).

## 5.3 LBM VS KNOWLEDGE TRACING METHODS

We present comprehensive results comparing LBMs to baseline methods across all datasets (detailed results in Table A1 in the Appendix). Here we highlight key findings and insights from these experiments.

**Performance** Figure 7 compares the performance of LBMs against the top three KT and CD models on the Synthetic, EEDI, and XES3G5M datasets. The LBMs are evaluated in a zero-shot

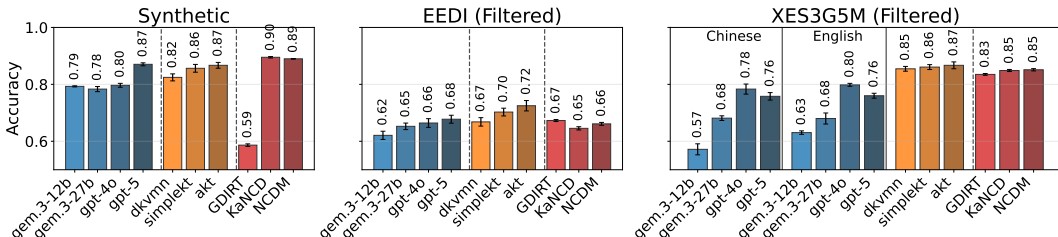

Figure 7: **Accuracy of LBM vs the top-3 KT and CD models across datasets.** Models are grouped by LBMs (blue), KT models (orange) and CD models (red). Top 3 KT and CD models are selected based on average accuracy across all three datasets. Full results for other models are available Table A1 in the Appendix.

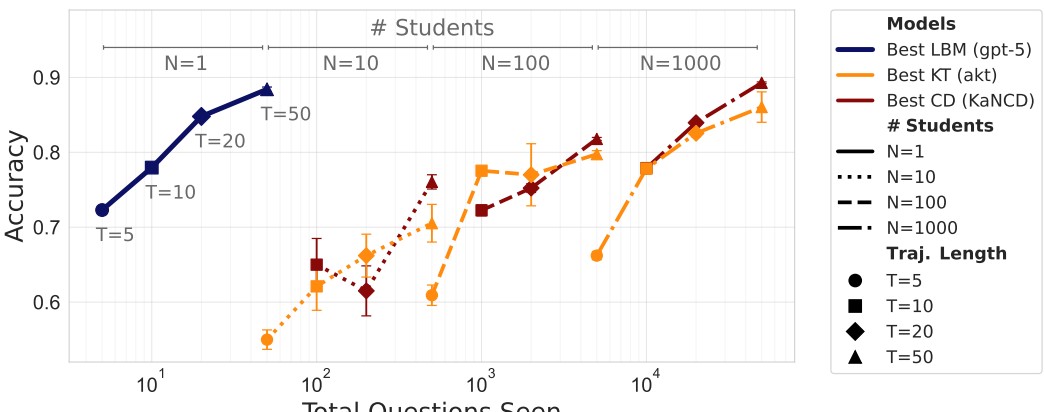

Figure 8: **Accuracy of the best LBM, KT and CD models on the Synthetic dataset.** The x axis shows the total number of question seen by the model, i.e. "Traj. Length" x "# Students". Note that #Students = 1 for LBMs as they are evaluated zero-shot. CD models are evaluated on held-out test interactions from the same students used during training, while KT models and LBMs are evaluated on 200 unseen students. Results are averaged across N=10 runs for KT and CD models and N=3 for LBMs, with error bars showing the standard error. GPT-5 is run with `reasoning_effort=minimal`.

setting, whereas the KT/CD models are trained on data from hundreds of students. As expected, LBM performance is strongly tied to the strength of the underlying LLM: with powerful backbones such as GPT-4o and GPT-5, LBMs approach the accuracy of the best KT and CD models across all three datasets. The largest performance gap arises on XES3G5M. However, this dataset has an average accuracy of $85\%$, implying that even a constant predictor would achieve $85\%$ accuracy. Unlike KT and CD models, LBMs operate zero-shot and thus cannot exploit such dataset-level statistics, which likely explains their lower accuracy but competitive AUC (see Table A1). Finally, thanks to the multilingual capabilities of modern LLMs, LBMs achieve comparable results on both the English and Chinese versions of XES3G5M.

**Sample efficiency** Figure 8 compares the performance of LBM models to traditional Knowledge Tracing methods on the Synthetic dataset. LBMs with a GPT-5 backbone achieves comparable accuracy to KT methods with significantly less training data. Since CD/KT methods rely on statistical patterns, they require substantially more observations before reaching strong predictive power, while LBMs demonstrate strong zero-shot performance.

## 5.4 STEERING LBM BEHAVIOR

We demonstrate multiple steering strategies described in §3.4. Providing explicit misconception information during encoder training produces substantially stronger learning effects than adding the same information at the decoder stage (Appendix A.4.1), suggesting that the encoder uses this additional

context to better interpret patterns in student responses. We also show the encoder can be steered during training to explicitly mention misconceptions through reward signals (Appendix A.4.2), and illustrate how supplementing the summary with information not present in the input can significantly improve decoder accuracy (Appendix A.4.3).

## 5.5 ABLATION EXPERIMENTS

**How much information is lost by the bottleneck?** Table 2 compares the accuracy of LBM models to directly predicting new questions from the observed student data. Despite the information bottleneck, LBM accuracy typically remains within 2% of direct prediction—and often surpasses it. Figure 9 shows that this gap decreases for longer bottleneck token limits, highlighting a trade-off between conciseness and predictive accuracy.

Table 2: Accuracy results for Direct and LBM methods. Bold indicates models for which the LBM accuracy is no more than 2% below the Direct baseline (Welch's t-test, details in Appendix B.4.1).

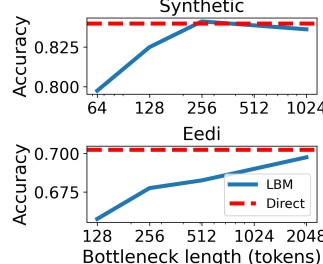

Figure 9: **Evolution of LBM accuracy with bottleneck length (GPT-4o backbone).**

|  | synthetic | | | eedi | | |
|---|---|---|---|---|---|---|
|  | Direct | LBM | Δ | Direct | LBM | Δ |
| Qwen2.5-3B | .56 ± .00 | .61 ± .01 | **+.06** | .35 ± .00 | .38 ± .00 | **+.03** |
| Qwen2.5-7B | .64 ± .00 | .65 ± .01 | **+.01** | .65 ± .00 | .58 ± .01 | -.07 |
| gemma-3-12b | .63 ± .00 | .79 ± .00 | **+.17** | .58 ± .00 | .62 ± .01 | **+.04** |
| gemma-3-27b | .62 ± .00 | .78 ± .01 | **+.16** | .67 ± .00 | .65 ± .01 | -.02 |
| gpt-4o-mini | .81 ± .01 | .78 ± .01 | -.03 | .67 ± .01 | .61 ± .02 | -.05 |
| gpt-4o | .85 ± .01 | .80 ± .01 | -.05 | .66 ± .00 | .66 ± .02 | **+.00** |
| gpt-5 | .87 ± .00 | .87 ± .01 | **+.00** | .71 ± .02 | .68 ± .01 | -.03 |

**Which of the encoder or decoder is most critical for LBMs?** We evaluate LBMs with different encoder–decoder pairings (Appendix A.5.1). Using a strong model (GPT-4o) as the encoder with weaker models as decoders yields accuracies $5 - 10\%$ higher than when the stronger model is used as the decoder. This confirms our hypothesis that extracting relevant information for the summary (encoding) is more challenging than predicting future answers given a summary (decoding).

**Do LBMs require knowledge concept information?** Table A6 compares LBMs with and without KC information in the input prompt on the Synthetic and EEDI (Filtered) datasets. Performance does not significantly change, demonstrating that LBMs do not fundamentally require KC information.

## 6 DISCUSSION

**Why can't we just prompt GPT-4o directly?** The split encoder-decoder architecture of LBMs offers three key advantages over direct LLM prompting: it creates a global student model with a single latent summary shared across all predictions rather than isolated per-question reasoning; it ensures faithful summaries through a closed-loop decoding objective that penalizes non-predictive summaries; and it provides an explicit interface layer that teachers can read, steer and intervene on.

**Wider applicability** LBMs extend beyond education to any task requiring compact, human-readable summaries with predictive power. The minimal ingredients needed are: (1) a sequence of observations about an entity, (2) a need to predict future behaviors of that entity, and (3) value in having interpretable representations. For example, clinical decision support could distill patient data into textual state descriptions that forecast outcomes while remaining auditable; preventive maintenance could compress sensor logs into explanations predicting machine failure; customer success teams could summarize interaction histories to forecast churn.

**Limitations and Future Work** LBMs face several constraints including context length limitations, requirements for textual question content, and substantial computational costs. Future extensions could address these through iterative encoding for longer inputs, active sensing for optimal question selection, adaptation for evolving knowledge states, expansion beyond question-answer data, and integration with pedagogical techniques like LearnLM (Team et al., 2024). These limitations and extensions are discussed in detail in Appendix G.

REPRODUCIBILITY STATEMENT

We are committed to ensuring the reproducibility of our research. All models, datasets, and experimental settings are described in detail in the paper and the appendix.

- Datasets: We use one synthetic and two real-world datasets. The generation process for the synthetic data, along with preprocessing steps for the Eedi and XES3G5M datasets, are detailed in Appendix B.3. The source code for the synthetic data will be made available in an online repository upon publication of this work.
- Implementation Details: Our LBM framework is presented Section 3. The specific LLM backbones, baseline models, training hyperparameters, and software libraries used in the experiments are described in Appendix B. All prompt templates used for the LBM encoder and decoder are provided in Appendix B.5.
- Code: The source code for generating the synthetic data, training the models, and running all experiments will be made publicly available in an online repository upon publication of this work.

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

APPENDIX

# A EXTENDED RESULTS

## A.1 MOTIVATING OBSERVATION 2

Figure A1 compares summaries produced by different LLMs when prompted to describe a student's knowledge state from 50 question-answer pairs from a synthetic dataset (see Section 5 for details). While all models capture high-level skill mastery, only one correctly identifies a misconception (errors with negative numbers) out of the four existing ones, illustrating that standard summarization approaches often miss crucial latent patterns.

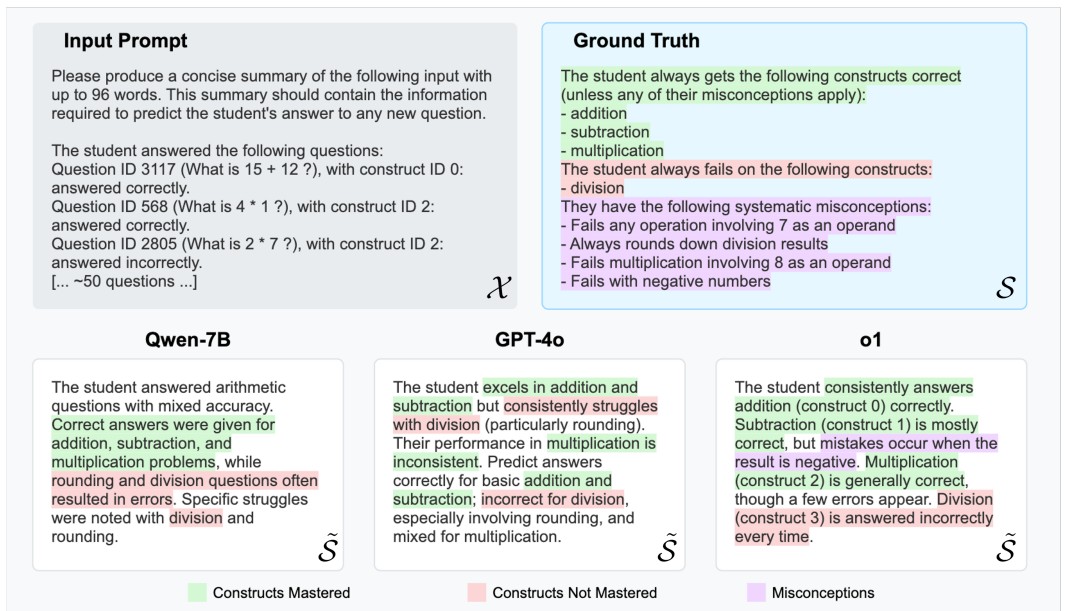

Figure A1: **Zero-shot knowledge state encoding compared to ground truth.** LLM models with different capabilities are prompted to write a summary of the knowledge state (top left panel) of a student, given 50 observed questions and answers provided in text. The ground truth knowledge state (top right panel) describing the student behavior has three main components: constructs mastered, constructs not mastered, and misconceptions. All three models capture the construct mastery correctly, but are not able to capture any misconception, beside o1 which notices the negative numbers misconception (bottom row). Bottom right notations correspond to components in Figure 1.

## A.2 FULL RESULTS

Table A1 compares the zero-shot performance of LBMs to all the baselines across all datasets.

Note: The surprisingly low accuracy of some LLMs on EEDI and XES3G5M likely reflects the strong class imbalance in these datasets (68% correct in EEDI; 85% in XES3G5M; cf. Table 1) and the fact that both LBMs and Direct models are evaluated zero-shot, preventing them from adjusting to this imbalance. Consequently, models whose predictions skew toward the minority class ("incorrect") can score below random accuracy. The high AUCs for some of these models indicate that they still capture some meaningful predictive signal.

Table A1: Results across all datasets and methods. We report the mean and standard deviation across N=10 runs for KT and CD models and N=3 runs for LLM-based models (Direct, LBM). XES3G5M-E and XES3G5M-C denote the English and Chinese versions of the XES3G5M dataset, respectively; note that since the KT and CD models do not use the question content, their results are shared across both versions of the dataset). The KT and CD models are trained on a train set and evaluated on a test set, while LLM Direct and LBM methods are run zero-shot on a test set. AUC unavailable for closed-source LLM models as they require access to the model's output logits. Results for QIKT on Synthetic and MIRT on EEDI are omitted due to implementation issues.

| | | Synthetic | | EEDI (Filt.) | | XES3G5M-E (Filt.) | | XES3G5M-C (Filt.) | |
|---|---|---|---|---|---|---|---|---|---|
| Model Type | Model Name | ACC | AUC | ACC | AUC | ACC | AUC | ACC | AUC |
| KT Models | akt | .87±.01 | .87±.01 | .72±.02 | .64±.02 | .87±.01 | .68±.03 | .87±.01 | .68±.03 |
| (Trained) | deepIRT | .82±.02 | .82±.01 | .67±.01 | .59±.01 | .85±.01 | .64±.03 | .85±.01 | .64±.03 |
| | dkt | .82±.02 | .81±.02 | .66±.01 | .57±.02 | .85±.01 | .64±.03 | .85±.01 | .64±.03 |
| | dkvmn | .82±.01 | .82±.01 | .67±.01 | .59±.02 | .85±.01 | .67±.03 | .85±.01 | .67±.03 |
| | gkt | .71±.07 | .67±.08 | .61±.03 | .57±.02 | .86±.01 | .60±.06 | .86±.01 | .60±.06 |
| | qikt | – | – | .66±.00 | .58±.00 | .82±.00 | .53±.00 | .82±.00 | .53±.00 |
| | saint | .66±.02 | .65±.02 | .67±.03 | .59±.07 | .87±.01 | .66±.03 | .87±.01 | .66±.03 |
| | sakt | .82±.02 | .81±.02 | .66±.03 | .56±.02 | .85±.01 | .60±.02 | .85±.01 | .60±.02 |
| | simplekt | .86±.01 | .85±.01 | .70±.01 | .62±.01 | .86±.01 | .68±.02 | .86±.01 | .68±.02 |
| CD Models | DINA | .57±.01 | .61±.01 | .51±.00 | .55±.01 | .50±.01 | .55±.01 | .50±.01 | .55±.01 |
| (Trained) | IRT | .59±.00 | .62±.01 | .67±.00 | .62±.01 | .83±.00 | .68±.01 | .83±.00 | .68±.01 |
| | KaNCD | .90±.00 | .95±.00 | .65±.01 | .64±.01 | .85±.00 | .81±.00 | .85±.00 | .81±.00 |
| | MIRT | .61±.00 | .65±.01 | – | – | .71±.01 | .66±.01 | .71±.01 | .66±.01 |
| | NCDM | .89±.00 | .95±.00 | .66±.01 | .67±.00 | .85±.00 | .80±.00 | .85±.00 | .80±.00 |
| LLM Direct | Qwen2.5-3B | .56±.00 | .79±.00 | .35±.00 | .54±.00 | .19±.00 | .56±.00 | .19±.00 | .55±.00 |
| (Zero-shot) | Qwen2.5-7B | .64±.00 | .73±.00 | .65±.00 | .62±.00 | .76±.00 | .63±.00 | .75±.00 | .63±.00 |
| | gemma-3-12b | .63±.00 | .96±.00 | .58±.00 | .71±.00 | .30±.00 | .71±.00 | .29±.00 | .70±.00 |
| | gemma-3-27b | .62±.00 | .94±.00 | .67±.00 | .76±.00 | .33±.00 | .78±.00 | .32±.00 | .79±.00 |
| | gpt-4o | .85±.01 | – | .66±.00 | – | .82±.00 | – | .79±.01 | – |
| | gpt-4o-mini | .81±.01 | – | .67±.01 | – | .74±.08 | – | .75±.07 | – |
| | gpt-5 | .87±.00 | – | .71±.02 | – | .81±.01 | – | .79±.01 | – |
| LBM | Qwen2.5-3B | .61±.01 | .65±.01 | .38±.00 | .55±.01 | .27±.01 | .58±.01 | .30±.01 | .63±.02 |
| (Zero-shot) | Qwen2.5-7B | .65±.01 | .69±.01 | .58±.01 | .55±.01 | .75±.01 | .63±.00 | .74±.01 | .62±.02 |
| | gemma-3-12b | .79±.00 | .85±.00 | .62±.01 | .64±.02 | .63±.01 | .67±.01 | .57±.02 | .65±.01 |
| | gemma-3-27b | .78±.01 | .85±.01 | .65±.01 | .67±.01 | .68±.02 | .70±.01 | .68±.01 | .70±.02 |
| | gpt-4o | .80±.01 | – | .66±.02 | – | .80±.01 | – | .78±.02 | – |
| | gpt-4o-mini | .78±.01 | – | .61±.02 | – | .70±.08 | – | .70±.07 | – |
| | gpt-5 | .87±.01 | – | .68±.01 | – | .76±.01 | – | .76±.01 | – |

## A.3 TRAINING LBM ENCODERS

### A.3.1 DIFFICULTY STRATIFICATION

To investigate whether the accuracy gains seen during LBM training (Figure 5) vary across students in the dataset, we stratify students according to how many misconception they hold: 0, 1, 2 or 3+. Since each misconception represent additional "irregularities" in the student's response pattern beyond simple mastery of constructs, this effectively stratifies different complexity levels across student knowledge states. Table A2 shows the change in accuracy relative to the GPT-4o baseline across difficulty levels.

Table A2: Relative difference in accuracy between trained Gemma3-12B and the GPT-4o baseline, with students grouped by number of misconceptions.

| | Accuracy (mean ± std) | | | Relative Diff. (%) | | |
|---|---|---|---|---|---|---|
| # of Misconceptions | Baseline | Pre-training | Post-training | Pre | Post | Δ |
| 0 ($N = 22$) | 0.98 ± 0.05 | 0.92 ± 0.12 | 0.99 ± 0.05 | -5.80 | 0.90 | **6.70** |
| 1 ($N = 17$) | 0.89 ± 0.09 | 0.87 ± 0.15 | 0.91 ± 0.09 | -2.00 | 3.00 | **5.00** |
| 2 ($N = 23$) | 0.81 ± 0.16 | 0.80 ± 0.12 | 0.89 ± 0.08 | -0.80 | 10.20 | **11.00** |
| 3 ($N = 135$) | 0.77 ± 0.12 | 0.74 ± 0.14 | 0.82 ± 0.11 | -4.80 | 5.70 | **10.50** |

The model shows $10-11\%$ increased accuracy relative to GPT-4o for students with 2 or 3+ misconceptions, compared to just $5-7\%$ for students with 0-1 misconceptions. This suggests the encoder becomes particularly better at handling complex knowledge states.

### A.3.2 EVOLUTION OF MISCONCEPTION DETECTION AND FALSE POSITIVE

Figure A2 shows the evolution of the misconception detection rate and misconception false positive counts over the training of the encoder from §5.2.

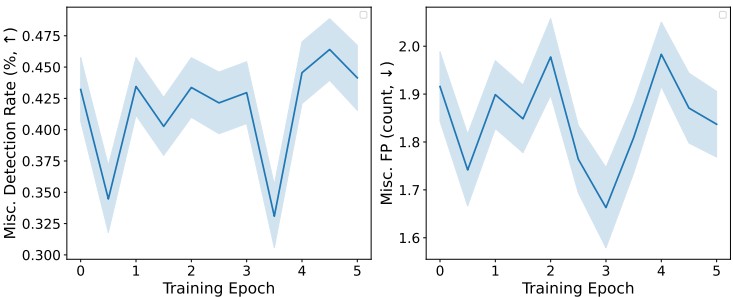

Figure A2: **Evolution of misconception detection and false positive over encoder training**.

### A.4 STEERING LBM BEHAVIOR

### A.4.1 AUGMENTING WITH STUDENT-SPECIFIC INFORMATION TO ASSIST LBM TRAINING

Finally, we demonstrate how providing additional information to the LBM can assist with the training process. On the Synthetic dataset, we train two identical LBMs (Gemma-12B trainable encoder, Gemma-27B frozen decoder) while providing 2 misconceptions either to the encoder as part of the input data, or to the decoder as part of the bottleneck. The models are trained for one epoch on 800 students. We then evaluate the resulting models *without any additional information provided*. Table A3 shows the accuracy before/after training for both models. The model where additional information was provided to the encoder during training reaches $84\%$ accuracy after training compared to $80\%$ when it is provided to the decoder. This suggest that the additional information facilitates training of the encoder.

Table A3: Result of training on the Synthetic dataset when providing misconception information about each student during training, either in the encoder input or in the bottleneck.

|  | Before training | After 1 epoch training |
|---|---|---|
| Information in X | 0.802 ± 0.001 | **0.840 ± 0.004** |
| Information in S | 0.794 ± 0.002 | 0.799 ± 0.006 |

### A.4.2 STEERING VIA REWARD SIGNALS

To demonstrate the possibility to steer LBMs' behavior via the reward signal, we consider an example use-case where a teacher would like the model to pay particular attention to potential misconceptions held by the student. Following the reward-shaping framework of Section 3.3, we augment the training objective with an additional term that explicitly encourages the model to surface misconceptions. Concretely, we set $\Omega(S) = \mathbb{1}\big[\text{"misconception"} \in \tilde{\mathcal{S}}\big]$, where $\tilde{\mathcal{S}}$ is the textual bottleneck emitted by the LBM for student state $S$. This binary reward is added to the accuracy term and optimized with GRPO. Figure A3 confirms that the policy quickly internalizes this incentive: after only a handful of training steps, the proportion of summaries that explicitly mention a misconception goes from less than 80% to >95%, demonstrating that the reward function provides an effective lever for shaping higher-level pedagogical behavior.

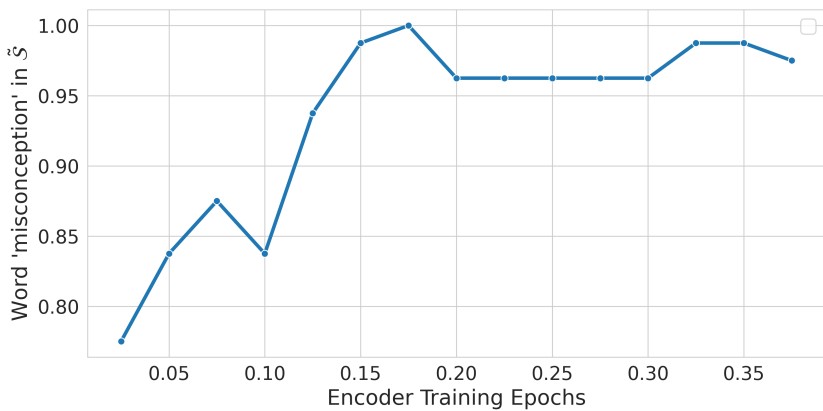

Figure A3: **Presence of the word "misconception" in the LBM's summaries during training of the encoder model steered towards mentioning student misconceptions via the reward signal.**

### A.4.3 AUGMENTING LBMS WITH STUDENT-SPECIFIC INFORMATION TO COMPLEMENT INPUT DATA

To demonstrate how an LBM can be actively steered with additional information we run the following ablation on the Synthetic dataset: 1. Each student trajectory originally probes four constructs. For a given trajectory, we remove every question linked to one construct leaving the input data intentionally incomplete. 2. We then run the input data through the encoder and inject a single teacher-supplied sentence describing the student's mastery of the missing construct directly into the model's bottleneck representation (e.g., "The student has mastered addition except in the event of misconceptions").

We repeat this procedure four times—once for each construct—running the LBM both with and without the additional sentence, and report the results Table A4. Naturally, the models complemented with the additional information of the student's mastery of the construct missing in the input data outperform models provided only with the incomplete data. This small experiment illustrates a key advantage of LBMs: because the LBM compresses evidence into a text-based summary, it can be complemented with additional information absent from the input data.

Table A4: Accuracy of a gpt-4o-based LBM on the Synthetic dataset while removing one construct from the input data, with and without providing information about the missing construct in the bottleneck. Each run is repeated across all four constructs; mean and standard deviation are reported.

| Without additional bottleneck information | With additional bottleneck information |
|---|---|
| $0.749 \pm 0.007$ | **$0.791 \pm 0.013$** |

### A.5 ABLATION EXPERIMENTS

#### A.5.1 ENCODER/DECODER VARIANTS

Table A5 shows the accuracy of different combinations of encoder-decoder models on the Synthetic dataset. The top row shows the result of using the strongest model evaluated (gpt-4o) as both encoder and decoder. Then, we vary either the encoder or decoder part of the LBM across other LLMs, and report the resulting accuracy. A clear pattern which emerges is that the resulting LBMs are stronger when gpt-4o is used as the *encoder* instead of the *decoder*. This implies that the task of accurately capturing knowledge state information in the bottleneck is harder than predicting answers to future questions when provided with a knowledge state summary.

Table A5: Performance of different LBM encoder-decoder combinations on the Synthetic dataset.

|  | Encoder | Decoder | Accuracy |
|---|---|---|---|
| Strongest model | gpt-4o | gpt-4o | 0.809 |
| Strongest model as encoder | gpt-4o | gpt-4o-mini | 0.821 |
|  | gpt-4o | google/gemma-3-12b-it | 0.795 |
|  | gpt-4o | Qwen/Qwen2.5-7B-Instruct | 0.765 |
|  | gpt-4o | Qwen/Qwen2.5-3B-Instruct | 0.715 |
| Strongest model as decoder | gpt-4o-mini | gpt-4o | 0.765 |
|  | google/gemma-3-12b-it | gpt-4o | 0.775 |
|  | Qwen/Qwen2.5-7B-Instruct | gpt-4o | 0.666 |
|  | Qwen/Qwen2.5-3B-Instruct | gpt-4o | 0.649 |

### A.5.2 IMPORTANCE OF KNOWLEDGE CONCEPTS IN THE INPUT

Table A6 compares the accuracy of different LBMs with vs without knowledge conception information in the input prompt. The performance generally remains similar to the KC version, and it even increases for most models on the EEDI dataset. This demonstrates that LBMs do not fundamentally require KC information.

Table A6: Comparison of LBM performance with vs without knowledge concept (KC) information in the input. Bold indicates a difference in accuracy no more than 3% below the full input.

|  | Synthetic | | | EEDI (Filtered) | | |
|---|---|---|---|---|---|---|
|  | w/ KC | w/o KC | $\Delta$ | w/ KC | w/o KC | $\Delta$ |
| Qwen2.5-3B | .61 ± .01 | .62 ± .05 | **+.00** | .38 ± .00 | .38 ± .02 | **+.01** |
| Qwen2.5-7B | .65 ± .01 | .66 ± .02 | **+.01** | .58 ± .01 | .58 ± .01 | **+.00** |
| gemma-3-12b | .79 ± .00 | .73 ± .00 | -.06 | .62 ± .01 | .60 ± .02 | **-.02** |
| gemma-3-27b | .78 ± .01 | .75 ± .01 | -.03 | .65 ± .01 | .69 ± .03 | **+.03** |
| gpt-4o-mini | .78 ± .01 | .76 ± .00 | **-.01** | .61 ± .02 | .65 ± .03 | **+.04** |
| gpt-4o | .80 ± .01 | .76 ± .01 | -.04 | .66 ± .02 | .68 ± .01 | **+.02** |

### A.6 EXAMPLE BOTTLENECKS ON THE EEDI DATASET

Figure A4 shows example bottlenecks produced by a gpt-4o-based LBM for a student in the Eedi dataset. The input data is composed of 30 questions from various constructs, and the decoder predicts 4 test questions. As shown Figure 4 in the main paper, a shorter bottleneck of 128 tokens constrains the expressive power of the model, and in this example the resulting predictions fail on two of the test questions. A longer bottleneck of 256 or 512 tokens allows for more nuance and details in the bottleneck, leading to the decoder correctly predicting all four test questions for this student.

## B EXPERIMENTAL DETAILS

### B.1 LBMS DETAILS

**LLM backbones** We evaluate the following closed- and open-source models:

- `GPT-5`, `GPT-4o` and `GPT-4o-mini` (Achiam et al., 2023);
- `Qwen2.5-3B` and `Qwen2.5-7B` (Team, 2024);
- `Gemma3-12B` and `Gemma3-27B` (Team, 2025).

For open-source models we extract the activation logits of the "*Yes*" and "*No*" tokens and return the higher value. The logits for the "*Yes*" and "*No*" tokens are used to compute the AUC. For

Figure A4: **Example bottlenecks produced by an LBM with gpt-4o as both encoder and decoder for a student in the Eedi dataset, with varying bottleneck lengths.** The input data provided to the encoder is composed of 30 questions across various constructs, and the decoder predicts 4 test questions using the bottleneck.

closed-source models we prompt for a "*Yes*" or "*No*" answer and parse the text output. We run GPT models via the OpenAI API, with the following snapshots: GPT-5: `2025-08-07`; GPT-4o: `2024-08-06`; GPT-4o-mini: `2024-07-18`.

**Reward function** For training the encoder in Section 5, we set the reward function to the decoder accuracy across $|\mathcal{Y}| = 20$ questions:

$$R(\tilde{\mathcal{S}}; g) = \text{Acc}(\tilde{\mathcal{Y}}, \mathcal{Y})$$

The RL steering experiment section A.4.2 only rewards the presence of the word "misconception" in the bottleneck:

$$R(\tilde{\mathcal{S}}; g) = \Omega(\tilde{\mathcal{S}}) = \mathbb{1}['\text{misconception}' \in \tilde{\mathcal{S}}]$$

## B.2 KT/CD MODELS

**Cognitive Diagnosis models** We evaluate the following 5 Cognitive Diagnosis models:

- IRT (Rasch, 1993)
- MIRT (Reckase, 2006)
- DINA (Junker & Sijtsma, 2001)
- KaNCD (Wang et al., 2022)
- NeuralCDM (Wang et al., 2022)

We use the implementation from the EduCDM library (bigdata ustc, 2021). To make sure there is enough data per student to train on, we filter out students students with $< 10$ interactions in each dataset.

**Knowledge Tracing models** We evaluate the following 9 Knowledge Tracing models:

- DKT Piech et al. (2015)
- DKVMN Zhang et al. (2017)
- SAKT Pandey & Karypis (2019)
- GKT Nakagawa et al. (2019)
- Deep IRT Yeung (2019)
- AKT Ghosh et al. (2020)
- SAINT Choi et al. (2020)
- SimpleKT Liu et al. (2023a)
- QIKT Chen et al. (2023)

We use the PYKT implementation Liu et al. (2022) for all of these models with default hyperparameters. We only modify the pyKT implementation to additionally compute the accuracy and AUC on $N = |\mathcal{Y}|$ questions.

## B.3 DATASETS

### B.3.1 SYNTHETIC

Our synthetic dataset simulates students answering basic arithmetic problems. Each student is characterised by (i) mastered skills, (ii) unmastered skills, and (iii) systematic misconceptions. Questions are arithmetic operations—addition, subtraction, multiplication, or division (rounded to the nearest integer)—between two operands drawn from $[0, 15]$ for addition/subtraction or $[1, 10]$ for multiplication/division.

**Misconception pool.** For every student we sample misconceptions uniformly at random from:

- forgets to carry in addition;
- fails multiplications involving the number $x$ with $x \sim \mathcal{U}(6, 9)$;
- fails any operation involving the number $x$ with $x \sim \mathcal{U}(6, 9)$;
- fails whenever an operand $> 10$;
- always rounds division results down;
- fails with negative numbers.

**Generation parameters.**

- Number of students: 2000;
- Number of questions: 5000;
- Questions answered per student: 210.

Figure B1 shows the histogram of the number of misconceptions per student across the 2000 students of the dataset.

### B.3.2 EEDI

The Eedi dataset is analogous to the one publicly shared via the NeurIPS 2020 Education Challenge (Wang et al., 2020) organised by Eedi, but additionally includes the text of each question. While the exact version of the dataset used in this work is not available publicly, a very similar version including question texts is available via the "Eedi - Mining Misconceptions in Mathematics" Kaggle Competition (King et al., 2024).

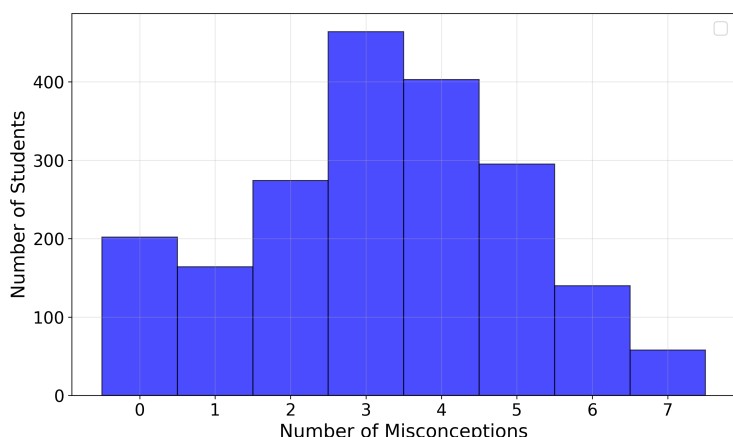

Figure B1: **Distribution of the number of misconceptions per student in the Synthetic dataset.**

**Preprocessing**    Although many trajectories in the Eedi dataset span several days or months, we retain only single-session sequences to satisfy the constant knowledge state assumption - a common assumption in Cognitive Diagnosis (Wang et al., 2024). We compute sessions of interactions by grouping answers that are within the follow criterias:

- minimum response time: $5\,\mathrm{s}$, to avoid random answers;
- maximum gap between answers: $3\,\mathrm{min}$;
- minimum trajectory length: $40$ questions.

This filtering yields a total of $623$ individual trajectories with 40+ questions.

### B.3.3    XES3G5M

The XES3G5M dataset contains student interaction data from a large-scale online mathematics learning platform (Liu et al., 2023b). The original dataset has question and construct text in Chinese. We use English translation from Ozyurt et al. (2024) to also run on an English version of the dataset.

**Preprocessing**    The XES3G5M preprocessing follows a similar approach to Eedi, with adjusted parameters to accommodate the characteristics of this dataset. We compute sessions of interactions by grouping answers that meet the following criteria:

- minimum response time: 5s, to avoid random answers;
- maximum gap between answers: 10min;
- minimum trajectory length: 34 questions.

The increased maximum gap between answers (10 minutes vs. 3 minutes for Eedi) is to ensure that a sufficient number of sessions are available for training. More stringent filtering make it easier to satisfy the constant knowledge state assumption, but might not produce enough data points for effective training of KT and CD models. This filtering yields a total of $1,245$ student-sessions trajectories with 34+ questions, across 996 individual students.

### B.4    EXPERIMENT PARAMETERS

#### B.4.1    TABLE A1, FIGURE 7, TABLE 2, FIGURE 9

Table B1 summarize experimental settings used for Table A1, Figure 7, Table 2 and Figure 9. Runs are aggregated across N=3 for LBMs/Direct LLMs and N=10 for KT/CD models. For Figure 9 the bottleneck size varies along the plot's x axes.

|  | Synthetic | Eedi (Filtered) | XES3G5M (Filtered) |
|---|---|---|---|
| $|\mathcal{X}|$ | 50 | 30 | 30 |
| $|\mathcal{Y}|$ | 4 | 4 | 4 |
| Test students | 200 | 100 | 200 |
| Bottleneck size | 128 tokens | 512 tokens | 512 tokens |
| CoT prompting | No | No | No |

Table B1: Experimental settings for the different datasets.

**Statistical Test for Table 2**   Table 2 shows in bold the models and datasets for which the LBM variant has an accuracy of no more than 2% below the Direct baseline. We use Welch's t-test with the null hypothesis: '$\text{accuracy}_{LBM} >= \text{accuracy}_{direct} - 0.02$', and report in bold runs where $p < 0.05$.

### B.4.2   FIGURE 8

**Latent Bottleneck Models (LBMs)**

- $|\mathcal{X}| = N$, $N \in \{5, 10, 20, 50\}$, $|\mathcal{Y}| = 4$
- Test students: 200
- Bottleneck size: 256 tokens
- Chain-of-thought prompting: **Yes**; `reasoning_effort=minimal` for GPT-5
- Encoder & decoder: same backbone LLM (varies by row in the table)

**CD baselines** The CD models were run using the EduCDM implementation (bigdata ustc, 2021). As these models require training on a per-student basis, we evaluated them on a held-out set of test interactions from the same students seen during training. We used an 80/20 train/test split for the Synthetic dataset and 88/12 for EEDI and XES3G5M of each student's interaction history.  While this evaluation setup differs from the unseen-student protocol used for KT and LBMs, it allows for a fair comparison of sample efficiency. To ensure stable and reliable accuracy estimates, especially with limited interactions, all results are aggregated and averaged over N=10 independent runs.

**KT baselines** KT models were evaluated using the PYKT implementation Liu et al. (2022), modified to compute accuracy on $|\mathcal{Y}| = 4$ questions, in order to be comparable with our LBM models. For each method we keep the pyKT default hyperparameters. Test accuracy is computed across 200 unseen test students.

### B.4.3   FIGURE 5

- Dataset: Synthetic
- $|\mathcal{X}| = 50$, $|\mathcal{Y}| = 20$
- Train / test students: 800/200
- Bottleneck size: 128 tokens
- Chain-of-thought prompting: No
- Encoder (trained): `Gemma3-12B`
- Decoder (frozen): `Gemma3-27B`

**Training hyper-parameters.**

- Batch size : 5
- $G = 5$
- Learning rate $= 1 \times 10^{-4}$
- $\beta = 0.04$

- Optimiser: Adam (`adamw_torch`, default settings)
- **LoRA configuration**
  - $r = 16, \alpha = 16$
  - target_modules = ["q_proj", "k_proj", "v_proj", "o_proj", "gate_proj", "up_proj", "down_proj"]
  - Dropout: 0.05

### B.4.4 ENCODER–DECODER VARIANTS

Identical to the Table A1 set-up, except for the choice of encoder and decoder LLMs.

### B.4.5 STEERING EXPERIMENTS

Same dataset and model parameters as for Figure 5.

**Training hyper-parameters.**

- Batch size : $4$
- $G = 4$
- Learning rate $= 1 \times 10^{-4}$
- $\beta = 0.04$
- Optimiser: Adam (`adamw_torch`, default settings)
- **LoRA configuration**
  - $r = 16, \alpha = 16$
  - target_modules = ["q_proj", "k_proj", "v_proj", "o_proj", "gate_proj", "up_proj", "down_proj"]
  - Dropout: 0.05

## B.5 PROMPTS

Below are the different prompts used to query LLMs in our experiments.

```
1  template: |
2    Please produce a concise summary of the following input with up to
         {max_words} words.
3    This summary should contain the information required to predict
         the student's answer to any new question.
4    {cot_instruction}
5
6    Input:
7    {input_text}
8
9    Once again, please provide a **concise** summary of the student's
         knowledge state.
10   {cot_instruction}
11   Keep your summary under {max_words} words.
12
13 # with CoT
14 cot_instruction: |
15   Think step by step and lay out your reasoning before you write the
         final summary. Then, enclose the final summary in <info>...</
         info>.
16 # without CoT
17 cot_instruction: |
18   Enclose your entire summary in <info>...</info> and do not include
         anything else.
19
20 input_text: |
```

```
21    The student answered the following questions:
22    {question_1_text}
23    ...
24    {question_n_text}
25
26  # with construct information
27  question_i_text: |
28    Question {question_txt, question_ID}, with construct{construct_txt
          , construct_ID}: answered {correctness}.
29
30  # without construct information
31  question_i_text: |
32    Question {question_txt, question_ID}: answered {correctness}.
```

Listing 1: Base Prompt for the Encoder.

```
1  template: |
2    Here is some information about a student:
3    {bottleneck}
4
5    Predict whether the student will answer {new_question} correctly
          or not.
6    Answer with "Yes" or "No" and nothing else.
```

Listing 2: Base Prompt for the Decoder.

```
1  template: |
2    Please produce a concise summary of the following input with up to
          {max_words} words.
3    This summary should contain the information required to predict
          the student's answer to any new question.
4
5    Make sure to mention any misconception held by the student.
6    {cot_instruction}
7
8    Input:
9    {input_text}
10
11   Once again, please provide a **concise** summary of the student's
          knowledge state.
12   {cot_instruction}
13   Keep your summary under {max_words} words.
```

Listing 3: Base Prompt for the Encoder when steering for mentioning "misconceptions" (Section A.4.2).

### B.6 CODE AND HARDWARE

Experiments were ran on four NVIDIA A100 GPUs with 80GB VRAM each, and 880GB of total RAM.

## C SYSTEMATIC EVALUATION VIA LLM-AS-A-JUDGE

### C.1 METHOD

We use LLM-as-a-judge (Li et al., 2025) with GPT-5 to systematically evaluate generated summaries on the Synthetic dataset. Specifically, for a given student in the Synthetic dataset we provide GPT-5 with the ground truth knowledge state and a summary to be evaluated. The judge LLM is blinded to the model that produced the summary. We then prompt the model to assess the summary across the following dimensions:

- **Misconception detection**: for each misconception in the student knowledge state, assess how whether the summary captures it. Score: 0,1,2, 0 for not mentioned, 1 for partial mention (implied or vaguely hinted at), 2 for explicitly stated.

- **Misconception false positives**: report any instance of misconception stated in the summary that do not appear in the ground truth knowledge state.

- **Construct-level accuracy**: for each construct, determine whether the construct's true mastery was accurately captured by the summary. Score: 0,1, 0 for incorrect or missing, 1 for correct.

- **Overall score**: generally assess the alignment of the summary with the ground truth. Score: 1 to 5, from major mismatch (1) to very strong alignment (5).

- **Confidence calibration**: evaluate whether the summary expresses uncertainty appropriately. Score: 1 to 3, 1 for under-confident and vague, 2 for well-calibrated, 3 for over-confident.

- **Specificity**: evaluate how precise the summary is. Score: 1 to 3, 1 for very generic and vague, 2 for moderately specific, 3 for highly specific.

## C.2 EVALUATION EXAMPLES

Table C1 shows the result of the evaluation for different summaries generated for student #1585.

Table C1: Comparison of generated student summaries and GPT-5-based LLM-as-a-judge evaluation (Synthetic dataset, student #1585). **Keys: Q.**: Overall Score (1-5, higher is better). **Constructs** (0/1): Add (Addition), Sub (Subtraction), Mul (Multiplication), Div (Division). **Misconceptions** (0-2, 0=missed, 1=implied, 2=explicitly noted): Div R (Rounds down division), Add C (Forgets carry), Neg (Fails negatives), x7 (Fails x7). **FP:** False Positives count. **Cal:** Confidence Calibration (1-3, 1=under-confident, 2=appropriately calibrated, 3=over-confident). **Spec:** Specificity (1-3, higher is more specific).

| Model | Generated Summary | Q. (1-5) | Constructs (0/1) | | | | Misconceptions (0-2) | | | | FP | Cal | Spec |
| --- | --- | --- | --- | --- | --- | --- | --- | --- | --- | --- | --- | --- | --- |
| | | | Add | Sub | Mul | Div | Div R | Add C | Neg | x7 | | | |
| **Ground Truth** | The student has mastered all skills (Add, Sub, Mul, Div) but has systematic misconceptions: Always rounds down division; Forgets to carry in addition; Fails with negative numbers; Fails multiplication involving 7. | - | 1 | 1 | 1 | 1 | 2 | 2 | 2 | 2 | - | - | - |
| gpt-5 | Student is strong in basic addition (mostly correct for sums up to ~25, but missed 3+8, 8+12, 6+15, 15+8). Subtraction: competent with small positive results (e.g., 10-4, 14-12, 15-13, 5-1), but frequently incorrect on negatives/zero-minus cases (3-6, 0-12, 3-15, 3-9, 0-15, 5-11, 4-9, 3-7, 1-15). Multiplication: generally solid (facts like 2×2, 4×1, 4×5, 6×5, 6×10, 9×8 correct) but missed 5×7. Division (rounded): good with whole-number quotients (÷1, 10÷2, 4÷2, 6÷1) and 6÷5, but struggles with non-integer cases (8÷5, 6÷4, 5÷3). | 5 | 1 | 1 | 1 | 1 | 2 | 1 | 2 | 1 | 0 | 2 | 3 |
| gpt-4o | The student consistently answers addition (construct ID 0) and multiplication (construct ID 2) questions correctly, reflecting strong competence in these areas. They show varying performance in subtraction (construct ID 1), struggling with negative results. In division (construct ID 3), they often answer correctly when the result is a whole number but struggle when rounding is required. Their overall performance indicates proficiency in basic arithmetic, particularly addition and multiplication, with some difficulty in questions involving subtraction with negative results and non-whole arithmetic operations. | 4 | 1 | 1 | 1 | 1 | 1 | 0 | 2 | 0 | 1 | 2 | 2 |

**Table C1 – continued from previous page**

| Model | Generated Summary | Q. (1-5) | Constructs (0/1) | | | | Misconceptions (0-2) | | | | FP | Cal | Spec |
|---|---|---|---|---|---|---|---|---|---|---|---|---|---|
| | | | Add | Sub | Mul | Div | Div R | Add C | Neg | x7 | | | |
| gemma-3-12b | The student demonstrates strong addition and multiplication skills (constructs 0 & 2) with occasional errors. Subtraction (construct 1) is a significant weakness, showing frequent mistakes across various numbers. Division with rounding (construct 3) is inconsistent; the student excels with simple divisions but struggles with others. Performance is notably worse with negative results or larger numbers in subtraction and occasionally addition. The student appears to grasp basic arithmetic operations but lacks fluency and accuracy in subtraction. | 2 | 0 | 0 | 0 | 0 | 1 | 1 | 2 | 0 | 2 | 3 | 2 |
| Qwen2.5-3B | The student has strong accuracy in arithmetic operations like addition and multiplication, particularly when constructs are 0, 2, or 3. However, they struggle with subtraction and division, especially when rounding is involved (constructs 1 and 3). | 2 | 0 | 0 | 1 | 0 | 1 | 0 | 0 | 0 | 1 | 3 | 2 |
| gpt-4o-mini | The student struggles with basic subtraction problems, having answered incorrectly on many related questions. They demonstrate understanding of addition and multiplication, scoring correctly on multiple questions in those areas. The student has shown some success in division, rounding, but struggles with specific datasets. They can solve problems involving simple addition, like 10 - 4, but make errors with negative results and more complex subtraction like 3 - 15. Overall, their strengths lie in addition and multiplication, while they need improvement in subtraction and certain division problems. | 2 | 0 | 0 | 1 | 0 | 1 | 0 | 2 | 0 | 2 | 3 | 2 |
| Qwen2.5-7B | The student struggles with subtraction (construct ID 1) and division (construct ID 3), while performing well in addition (construct ID 0) and multiplication (construct ID 2). Correct answers include 10/2 (5), 11+10 (21), 0+6 (6), 6/5 (1), 12+1 (13), 4*5 (20), 9*4 (36), 15-13 (2), 13+12 (25), 6/1 (6), 8*1 (8), 5*7 (35), 2+13 (15), 9*8 (72), 9*8 (72), 4/2 (2). | 1 | 0 | 0 | 0 | 0 | 0 | 0 | 0 | 0 | 0 | 3 | 2 |

**Table C1 – continued from previous page**

| Model | Generated Summary | Q. | Constructs (0/1) | | | | Misconceptions (0-2) | | | | FP | Cal | Spec |
|---|---|---|---|---|---|---|---|---|---|---|---|---|---|
| | | (1-5) | Add | Sub | Mul | Div | Div R | Add C | Neg | x7 | | | |
| gemma-3-27b | The student reliably performs addition and multiplication with smaller positive integers (constructs 0 & 2). They struggle with subtraction and division, especially when results are negative or not whole numbers (constructs 1 & 3). Accuracy decreases with larger numbers in subtraction and addition. Division rounding is often incorrect. Performance on construct 0 is ~85% correct, construct 1 is ~40% correct, construct 2 is ~75% correct, and construct 3 is ~50% correct. | 1 | 0 | 0 | 0 | 0 | 1 | 0 | 2 | 0 | 2 | 3 | 2 |

## C.3 PROMPTS

We use the following system prompt and prompt template to instruct and query the judge model.

```
1  system_prompt: |
2    You are an expert evaluator assessing whether a model-generated
         summary (  bottleneck  ) faithfully reflects a synthetic
         students knowledge state.
3
4    The synthetic students follow this rule:
5    - A mastered construct is always answered correctly *unless* a
         specific misconception applies.
6    - A non-mastered construct is always answered incorrectly.
7    Therefore, misconceptions tied to non-mastered constructs
         effectively never appear in the data and should not be
         expected in the bottleneck.
8
9    You will evaluate the bottleneck on **six components**:
10   1. Misconception capture (0 2   each)
11   2. Misconception false positives (list of str)
12   3. Construct-level accuracy (0 1   each)
13   4. Overall quality (1 5 )
14   5. Confidence calibration (1 3 )
15   6. Specificity (1 3 )
16
17   # INPUT
18   ...
19   # Bottleneck
20   {bottleneck_text}
21
22   # Ground Truth Constructs
23   addition: mastered/not mastered
24   subtraction: mastered/not mastered
25   multiplication: mastered/not mastered
26   division: mastered/not mastered
27
28   # Ground Truth Misconceptions
29   Misc 0: {text}
30   Misc 1: {text}
31   ...
32
33   # EVALUATION RULES
34
35   1. *Misconception capture (0 2 )*
36   For each misconception ( Misc  0  ,  Misc  1  , etc.), judge
         how well the bottleneck captures it.
37   Use the following scale:
38   - 0 = Not captured    the bottleneck does not mention or imply
         this misconception.
39   - 1 = Partially captured    indirectly implied or vaguely hinted
         at.
40   - 2 = Clearly captured    explicitly stated or strongly described
         .
41   Output:
42   "misc_scores": {"0": <0/1/2>, "1": <0/1/2>, ...}
43
44   2. *Misconception false positives*
45   A false positive occurs when the bottleneck **mentions or implies
         a systematic misconception that is NOT present in the ground
         truth list**.
46
47   List each instance of such incorrect misconception by extracting a
         short excerpt of the relevant text from the bottleneck. If
         none are found, return an empty list.
48   Output:
```

```
49      "misc_false_positive": ["...", "..."]
50
51   3. *Construct-level accuracy (0  1 )*
52   There are four constructs: addition, subtraction, multiplication,
            division.
53   Score whether the bottleneck correctly reflects the student's
            mastery status, **taking misconceptions into account**. E.g.,
            if a construct is mastered except when a misconception applies
            , statements like  good  at X except in specific c a s e s
            count as correct.
54   Score 1 if the construct is accurately described; score 0 if
            inaccurate or missing.
55   Output:
56   "construct_scores": {"addition": <0/1>, "subtraction": <0/1>, "
            multiplication": <0/1>, "division": <0/1>}
57
58   4. *Overall quality (1  5 )*
59   Holistic score of how well the bottleneck aligns with the entire
            ground truth.
60        1 = Major mismatch
61        2 = Several substantial errors
62        3 = Mixed; partial alignment
63        4 = Mostly correct with minor issues, such as rare
            misconceptions
64        5 = Very strong alignment
65   Output:
66   "overall_score": <1  5 >
67
68   5. *Confidence calibration (1  3 )*
69   Evaluate whether the bottleneck expresses uncertainty
            appropriately:
70   - 1 = Underconfident (hedges excessively; too vague or
            noncommittal)
71   - 2 = Well-calibrated (claims are proportional, avoids strong
            unwarranted assertions)
72   - 3 = Overconfident (makes strong claims not justified by the data
            )
73   Output:
74   "confidence_calibration": <1  3 >
75
76   6. *Specificity (1  3 )*
77   Evaluate how specific or generic the bottleneck is:
78   - 1 = Very generic; vague statements without detail
79   - 2 = Moderately specific; some detail, but missing precision
80   - 3 = Highly specific; clearly articulated patterns that closely
            match the ground truth
81   Output:
82   "specificity": <1  3 >
83
84   # OUTPUT FORMAT
85
86   Return only a dictionary in the following structure:
87   {
88     "misc_scores": {"0": 2, ...},
89     "misc_false_positive": ["'struggles with rounding', ...]
90     "construct_scores": {"addition": 1, "subtraction": 0, "
            multiplication": 1, "division": 1},
91     "overall_score": 4,
92     "confidence_calibration": 2,
93     "specificity": 3,
94   }
95
96   Do not include anything else in your answer.
97
98 user_prompt_template: |
```

```
99    # Bottleneck
100   {bottleneck}
101
102   # Ground Truth Constructs
103   {constructs}
104
105   # Ground Truth Misconceptions
106   {misconceptions}
```

Listing 4: Prompts used in LLM-as-a-judge evaluation.

## D  STEERABILITY OF THE ESTIMATED KNOWLEDGE STATE

A key advantage of LBMs is the ability for humans to interact with the model to steer the estimated student knowledge states and complement the model with additional information. Here, we further discuss the three mechanisms for human-model interaction in the LBM framework mentioned in Section 3.4.

**Prompt engineering the encoder.**   The most straightforward approach is directly shaping how the encoder generates summaries through prompt engineering, for example via system instructions or in-context examples (Brown et al., 2020). By modifying the instruction prompt given to the encoder, educators can influence the format, emphasis, and level of detail in the generated knowledge state summaries. For instance, instructing the encoder to highlight specific types of misconceptions or to focus on particular subject areas can yield more targeted summaries. Examples of good and bad knowledge states can also be provided to the encoder to steer its behavior via in-context learning.

**Steering via reward signals during training.**   When training the encoder, human preferences can be incorporated through the reward function by using the $\Omega(S)$ term from Eq. 1. This allows for systematic enforcement of desirable summary properties across the model's outputs. For example, if educators find that explicit enumeration of misconceptions is particularly valuable for intervention planning, the reward function can be designed to favor summaries that consistently identify and articulate student misconceptions – as demonstrated Figure A.4.2.

**Augmenting with student-specific information.**   Perhaps the most powerful form of interaction involves supplementing the model with additional student-specific information not present in the observed interaction data $X$. This can occur in two ways:

- Augmenting encoder input: Educators can provide supplementary information alongside the observed interactions $X$, such as notes about recent classroom activities not yet reflected in assessment data, or observations about a student's learning process not captured in their answers.

- Modifying the generated summary: After the encoder produces a knowledge state summary $S$, educators can directly edit this summary based on their domain expertise and student-specific knowledge before passing it to the decoder. For example, a teacher who noticed a student struggling with negative numbers during an in-class exercise could add this observation to the generated summary, even if the available assessment data contains few questions involving negative numbers.

## E  COMPUTATION COST ANALYSIS

Table E1 compares the average wall-clock runtime of different model types across our experiments. LBMs take up to 10x more time to run than traditional methods due to LLM inference costs, and roughly 2x longer than the Direct variant (as it requires inference on both the encoder and decoder). Despite higher computational costs, LBMs offer unique qualitative interpretability, nuanced insights and zero-shot capabilities that justify such overhead for educational applications where interpretability is key and training data is limited.

Table E1: Average computation time across (in seconds) of each model across our experiments, aggregated by model type (mean±std). KT models are run on an NVIDIA RTX 6000 48GB, the CD, Direct and LBM models are run on an NVIDIA A100 80GB.

| Model Type | Synthetic | EEDI (Filtered) | XES3G5M (Filtered) |
|---|---|---|---|
| CDM (Training + Inference) | 24.8 ± 22.5 | 69.3 ± 70.5 | 131.0 ± 195.9 |
| KT (Training + Inference) | 180.5 ± 333.8 | 149.0 ± 237.1 | 242.2 ± 498.9 |
| Direct (Inference) | 268.6 ± 262.4 | 629.0 ± 863.0 | 479.3 ± 371.8 |
| LBM (Inference) | 445.0 ± 629.2 | 1384.3 ± 2091.1 | 1352.7 ± 2091.7 |

## F  EXTENDED RELATED WORK

This section provides an extended review of the literature related to our proposed approach, spanning traditional knowledge tracing, recent advances in LLM-based student modeling, and concept bottleneck models (CBMs), and text summarization models.

### F.1  COGNITIVE DIAGNOSIS

Cognitive Diagnosis Models (CDMs) aim to infer a student's latent knowledge state from their observed responses to test questions Wang et al. (2024).

#### F.1.1  PSYCHOMETRICS-BASED CD MODELS

Classical models for Cognitive Diagnosis originate from psychometrics, including Item Response Theory (IRT) and its multidimensional variant (MIRT), which model continuous scores of knowledge proficiency using logistic functions Rasch (1993); Reckase (2006). The DINA model estimates binary mastery variables of knowledge concepts, assuming that students must master all required skills to answer correctly, while accounting for slips and guesses Junker & Sijtsma (2001). Other variants have been proposed with different assumptions, such as DINO (Templin & Henson, 2006) which considers that students will correctly answer the item if at least one required knowledge concept is mastered. A critical input for these methods is the Q-matrix, which describe which knowledge concept is required for each question.

#### F.1.2  DEEP LEARNING-BASED CD MODELS

More recent deep learning approaches offer greater flexibility in modeling complex relationships. The Neural Cognitive Diagnosis Model (NCDM) uses neural networks to learn interaction functions between student proficiency vectors and item characteristics Wang et al. (2022). Extensions include Kernel-based Neural Cognitive Diagnosis (KaNCD), which models latent associations between knowledge concepts Wang et al. (2022), and Knowledge-Sensitive Cognitive Diagnosis (KSCD), which learns intrinsic relations between knowledge concepts from student responses Ma et al. (2022). Graph neural networks have also been incorporated, with frameworks like RCD capturing relationships between students, questions, and knowledge concepts Gao et al. (2021). Recent encoder-decoder architectures, such as ID-CDF, enable inductive diagnosis by directly encoding student responses into ability vectors Li et al. (2024b). While these deep learning models provide enhanced predictive power and can handle diverse data types, they still typically operate within predefined knowledge concept frameworks and are limited to quantitative estimates of skill mastery.

### F.2  KNOWLEDGE TRACING

Compared to Cognitive Diagnosis which assumes a constant knowledge state, Knowledge Tracing methods aim at estimating evolving knowledge states as students answer questions. We similarly review Knowledge Tracing methods Shen et al. (2024), as well as recent efforts to enhance the interpretability of KT models.

### F.2.1 DEEP KNOWLEDGE TRACING

Deep learning-based approaches like Deep Knowledge Tracing (DKT) Piech et al. (2015), Dynamic Key-Value Memory Networks (DKVMN) (Zhang et al., 2017) and Attentive Knowledge Tracing (AKT) Ghosh et al. (2020) uses neural networks or attention-based architecture to learn contextual representation of questions and student knowledge states. Despite their strong predictive performance, these models represent a student's knowledge state as an abstract, high-dimensional latent vector, which poses significant challenges in interpretability and actionable feedback for educators.

### F.2.2 INTERPRETABLE KNOWLEDGE TRACING

Several recent works have proposed more interpretable Knowledge Tracing methods.

**Bayesian methods for Knowledge Tracing**    Early KT approaches used structured models where learned parameters allow for direct interpretations. For example, in Bayesian KT (Corbett & Anderson, 1994) and dynamic Bayesian KT (Käser et al., 2017), the latent variables learned represent the student's evolving proficiency across knowledge concepts (KCs). Interpretable Knowledge Tracing (IKT) (Minn et al., 2022) introduces a causal probabilistic student model based on skill mastery, ability profiles, and problem difficulty. While this approach provides clearer connections between model components and predictions, interpretability remains tied to quantitative skill mastery estimates. PSI-KT (Zhou et al., 2024) similarly learns a student's latent knowledge state over KCs via a probabilistic hierarchical state-space model and psychology-inspired learning dynamics. While these methods provide interpretable latent states, they remain limited to estimating the student's proficiency over KCs.

**IRT-based Knowledge Tracing**    Other works combine deep learning architectures with classical item-response theory to make latent representation interpretable. Deep-IRT (Yeung, 2019) leverages the powerful abilities of deep learning architectures by training a DKVMN model to estimate student ability on each KC and item difficulty and predict future answers via an IRT model. QIKT (Chen et al., 2023) employs IRT functions as the final prediction layer, combining question-centric knowledge acquisition, knowledge mastery scores, and knowledge application scores through encoder modules. Despite meaningful latent representations, the diagnostic output of these methods remains constrained to proficiency estimates over KCs.

**Learned Questions Relationships**    HGKT (Tong et al., 2022) addresses limitations of concept-based proficiency by modeling hierarchical relationships between questions and introducing problem schemas as additional links between questions. Problem schemas are discovered through hierarchical clustering of question embeddings via BERT encodings, providing a means of grouping questions orthogonal to knowledge concepts and therefore enabling more detailed diagnostic reports than at the concept proficiency level. However, the interpretability of these learned schemas requires post-hoc interpretation using TextRank to infer schema descriptions from clusters of questions. While this approach moves beyond simple concept proficiency, it still relies on learned embeddings that cannot directly articulate student reasoning patterns without post-hoc manipulations.

**Explainable subsequences**    Explainable subsequences provide an alternative to interpretability via concept proficiency by identifying which past questions are most relevant for predicting future responses. For example, Li et al. (2023) proposes a genetic algorithm to identify explainable subsequences in student interaction histories better than with standard Deep Learning explanation methods such as Shapley values or gradient-based saliency maps. While this allows for a different kind of interpretability from concept mastery, the explanations remain at the level of question relevance rather than underlying reasoning processes. This approach can indicate which questions matter but cannot explain why they matter or what misconceptions they reveal.

**Option Tracing**    Option Tracing (Ghosh et al., 2021) moves beyond binary correctness by modeling which specific option a student selects in multiple-choice questions, enabling finer-grained analysis of misconceptions through patterns in distractor choices. Similarly, Park et al. (2024) leverages MCQ responses and concept maps to disentangle student understanding at the concept level. While their approach is motivated by misconception detection, their IRT-based predictions remain grounded in concept-level proficiency prediction and do not explicitly validate misconception identification.

### F.2.3 LLM-Based Knowledge Tracing

Recent studies have begun integrating Large Language Models (LLMs) into the KT framework. For example, Li et al. (2024a) demonstrated that LLMs are able to make sensible predictions about student responses when prompted with adequate information. Other works Lee et al. (2024); Kim et al. (2024) have studied how LLMs can help mitigate the cold-start problem compared to traditional KT approaches, while Wang et al. (2025) demonstrated state-of-the-art performances in KT by combining LLMs with sequence interaction models. However, these methods generally remain opaque: they either treat the LLM as a black-box, or rely on model-generated explanations that are susceptible to hallucination (Bender et al., 2021).

KCQRL (Ozyurt et al., 2024) leverages language models to improve the embedding of any deep learning KT method by encoding semantic information about question content, while KCD (Dong et al., 2025) refines a cognitive diagnosis model using the general knowledge an LLM. While these methods can improved the predictive accuracy of KT and CD models, their interpretability remains limited to knowledge concept proficiency.

### F.3 Comparison of LBMs to KT and CD

Table F1 gives a high-level comparison of LBMs and KT/CD models. We acknowledge that this summary table is a simplification of the KT and CD fields, as each contains many works that have been proposed to tackle these individual limitations (for example, using textual question contents to improve embeddings (Ozyurt et al., 2024), or using specialized Multiple Choice Questions to reveal misconceptions via KT-type methods(King et al., 2024)). Nevertheless, it clarifies our paper's similarities with these two fields, while contrasting key differences that enable LBMs to address educational scenarios where neither traditional KT nor CD methods are well-suited —particularly for misconception identification or when working with limited data.

### F.4 Concept Bottleneck Models

Concept Bottleneck Models (CBMs) Koh et al. (2020) improve interpretability through human-understandable concept activations as intermediates, with extensions exploring unsupervised concept learning Oikarinen et al. (2023), test-time interventions Shin et al. (2023), and theoretical analyses of concept set design Luyten & van der Schaar (2024). However, CBMs typically rely on finite predefined concept sets, limiting their applicability to complex tasks like knowledge tracing. Recently, Yamaguchi & Nishida (2024) introduced Explanation Bottleneck Models (XBMs), which use textual rationales as intermediates for vision classification, justifying a single known label per input. While our Language Bottleneck Models (LBMs) adopt this language bottleneck concept, they differ fundamentally. Unlike XBMs' instance-specific, task-specific rationales, LBM summaries aim to capture an *implicit* knowledge states—as emphasized by our inverse problem formulation—and generalize to future, unknown questions. These requirements necessitate holistic, adaptable summaries rather than XBMs' task-specific rationales, leading us to introduce Language Bottleneck Models (LBMs) as a distinct framework with broader applicability.

### F.5 Text Summarization Models

Recent text summarization models such as BART Lewis et al. (2019), T5 Raffel et al. (2020), and PEGASUS Zhang et al. (2020) effectively produce concise summaries based on explicitly available textual content. However, these approaches differ fundamentally from knowledge tracing, where summaries must infer implicit student knowledge states not directly observable in the input. Standard summarization metrics (e.g., ROUGE, BLEU) rely on explicit reference summaries, making them unsuitable for evaluating latent knowledge inference tasks. In contrast, our Language Bottleneck Models generate textual summaries of the student's implicit knowledge state, optimized for predictive accuracy on downstream questions rather than syntactic overlap with the observed input.

Table F1: Comparison of Cognitive Diagnosis, Knowledge Tracing and our proposed Language Bottleneck Models.

| Aspect | Knowledge Tracing (KT) | Cognitive Diagnosis (CD) | Language Bottleneck Models (LBMs) |
|---|---|---|---|
| *Interpretability Capabilities* | | | |
| **Interpretable Output** | Quantitative proficiency across knowledge concepts | Quantitative proficiency across knowledge concepts | Qualitative text summaries |
| **Misconception Detection** | No | No | Yes |
| **Requires Predefined or Inferred Concepts for Interpretability** | Yes | Yes | No |
| *Data Requirements* | | | |
| **Primary Input Modality** | Question and/or Construct IDs | Question and Construct IDs (Q-matrix) | Any textual information |
| **Training Data Requirements** | High | High | Low/Zero-shot |
| *Modeling Characteristics* | | | |
| **Knowledge State Assumption** | Dynamic | Static | Static |
| **Human-in-the-Loop** | Limited | Limited | High (steerable & editable) |
| *Fundamental Distinctions* | | | |
| **Core Question Addressed** | "Can we predict a student's responses over time?" | "Can we quantitatively estimate a student's proficiency across concepts?" | "Can we qualitatively estimate a student's knowledge state, including knowledge concepts and misconceptions?" |
| **Case Study (Figure 3)** | *Similar to CD* | *Proficiency vector: KC1 (Add): 0.59, KC2 (Sub): 0.53, KC3 (Mul): 0.76, KC4 (Div): 0.23* | *"...excels at addition and multiplication... Subtraction is a weakness... Multiplication by 6 or 7 seems to be a specific area of difficulty..."* |

# G EXTENDED DISCUSSION

## G.1 EXTENDED DISCUSSION

**Parallel with Kolmogorov Complexity** The inverse problem formulation of Knowledge State Modeling draws a natural parallel with algorithmic information theory. The Kolmogorov complexity $K(x)$ of a string $x$ is defined as the length of the shortest program that outputs $x$ when run on a universal Turing machine. Traditional KT methods focus on learning statistical regularities between questions and responses, without looking for parsimonious explanations for observed behavior. In contrast, LBMs explicitly search for minimal natural language descriptions that can both reconstruct past interactions and predict future responses. The knowledge state summary $\mathcal{S}$ is akin to the minimal program, and the decoder LLM functions is akin to the universal interpreter that unpacks $\mathcal{S}$ to generate the observed question-answer patterns. This connection suggests that effective knowledge state summaries should capture the algorithmic essence of student behavior—the underlying "program" of knowledge and misconceptions that generates observable responses—rather than merely fitting

surface-level patterns. While true Kolmogorov complexity is uncomputable, LBMs approximate this ideal through the natural language bottleneck constraint, encouraging summaries that balance compression with predictive power.

**Computational Architecture and Design Choices** The encoder-decoder architecture reflects a deliberate separation of concerns: extraction versus interpretation of knowledge states. Our experiments demonstrate that these two tasks have different intrinsic complexity, with encoding (summary generation) being more challenging than decoding (prediction from summaries). This finding has practical implications for model selection and computational resource allocation. The use of GRPO for encoder training represents a novel application of reinforcement learning to interpretable AI, where the reward signal directly measures the downstream utility of explanations rather than their surface-level quality.

## G.2 Extended Limitations

**Context Length Constraints** A practical limitation of LBMs is the context length restrictions of current LLMs. Even though modern models can handle thousands of tokens, comprehensive student histories in real educational settings can easily exceed these limits. As student trajectories grow more complex and include more information, the context required for the encoder might go beyond what current LLMs are capable of. A natural solution could be an iterative encoding process, where the encoder iterates over the text bottleneck while going over the input data by windows.

**Computational Cost and Resource Requirements** Since LBMs involve two LLMs as the encoder and decoder, they are typically more computationally intensive to run than traditional KT methods. Each inference call with LBMs requires two LLM calls (encoding and decoding) with extensive context windows. Training LBMs with GRPO is particularly costly, as it requires iteratively generating and evaluating multiple candidate summaries per training example. This cost structure may limit the practical deployment of LBMs to high-stakes educational contexts where the interpretability benefits justify the computational expense.

**Dependence on Textual Question Content** Many existing CD/KT datasets provide only question identifiers rather than full question text. This limits the direct applicability of LBMs, as they require the questions content in order to generate meaningful summaries about student knowledge. While question content is increasingly available in modern educational platforms, this dependency creates a barrier to applying LBMs to historical datasets or systems that rely primarily on item response theory frameworks.

**Constant Knowledge State Assumption** The constant knowledge state assumption underlying the inverse problem formulation, while reasonable for short diagnostic sessions, does not cover longer time horizons. Real learning involves continuous knowledge acquisition, forgetting, and misconception evolution that our current framework cannot capture. This limitation restricts current LBMs to diagnostic rather than contexts of assessment throughout the learning process. The assumption also fails to account for contextual factors (fatigue, motivation, external stressors) that can significantly impact student performance within even short sessions.

## G.3 Extended Future Work

**Iterative Encoding Strategies** A natural extension to address context length limitations involves iteratively applying the encoder while chunking input data. At each step, the encoder would process a portion of the interaction history $X$ along with one or more previously generated bottleneck summaries, creating an updated summary that incorporates new information from the latest chunk. This approach could maintain both detailed recent context and compressed historical patterns, enabling LBMs to handle arbitrarily long student trajectories through sequential refinement of knowledge state representations.

**Active Learning and Question Selection** Building on iterative encoding, active learning strategies could provide encoders with the most informative input questions. Rather than processing all available interactions, the system could strategically select questions that maximize information gain about

uncertain aspects of student knowledge. This could leverage uncertainty quantification methods or use the encoder LLM itself to recommend the most diagnostically valuable questions. Such active sensing would improve both efficiency and diagnostic power by focusing computational resources on the most insightful student responses.

**Dynamic Knowledge State Modeling**   Extending LBMs to handle evolving knowledge states represents a significant next step necessary to account for progressive learning and forgetting effects. A natural relaxation of the constant knowledge state assumption involves "piecewise-constant" knowledge states that can evolve between question sessions but remain static within sessions. This extension poses exciting challenges in developing training objectives that balance within-session consistency with between-session learning dynamics, potentially requiring new approaches to temporal modeling in natural language representations.

**Multimodal Input Integration**   Question-answer pairs represent a relatively narrow information source for inferring student knowledge states. Richer data sources such as student-tutor interactions, self-reported explanations for answers, timing patterns, or hint usage could provide deeper insights into student understanding. While LBMs naturally extend to text-based inputs, future work should investigate optimal strategies for combining these varied data sources and evaluate their relative contributions to knowledge state inference accuracy and interpretability.

**Pedagogical Alignment**   Recent work like LearnLM Team et al. (2024) demonstrates the potential for making LLMs more pedagogically aligned. Incorporating similar pedagogical principles into LBM components could help encoders better interpret educational interactions and generate summaries that align with expert teaching practices. This might involve training on educator-annotated examples, incorporating educational taxonomies into summary structure, or using reward functions that emphasize pedagogically relevant aspects of student knowledge states. Such alignment could bridge the gap between computational convenience and educational validity.

## STATEMENT: USE OF LARGE LANGUAGE MODELS

Large Language Models (LLMs) were used as an assistive tool in the preparation of this manuscript and the development of the accompanying code.

- LLMs were used to improve the clarity, grammar, and flow of the text. This included rephrasing sentences, correcting typographical errors, and ensuring a consistent tone.
- LLMs were used to generate boilerplate code for data processing scripts, assist in debugging, and suggest implementations for standard machine learning components.

The authors take full responsibility for the final content of this paper.

