# OpenReview forum: "Language Bottleneck Models: A Framework for Qualitative Cognitive Diagnosis"
_ICLR.cc/2026/Conference — Submitted to ICLR 2026_

### Official Review · Reviewer_aM4W · 2025-10-29

**Soundness:** 2
**Presentation:** 2
**Contribution:** 1
**Rating:** 2
**Confidence:** 4

**Summary:**

The paper proposes Language Bottleneck Models (LBMs) for cognitive diagnosis.
An encoder LLM compresses a student’s interaction history into a free-form text summary of their knowledge state; a decoder LLM then uses only this summary to predict future performance. This work instantiates LBMs, evaluate zero-shot and with RL (GRPO) fine-tuning of the encoder using decoder accuracy as reward, and report competitive accuracy vs. strong baselines on a synthetic arithmetic benchmark and two real datasets.

**Strengths:**

- Casting knowledge-state modeling as a language bottleneck is interesting and novel, different from fixed concept spaces in CD and learning embeddings in KT. The diagram on page 2 (Fig. 1) clearly positions LBMs against CD/KT and motivates the bottlenecked interface.
- The design choice to freeze the decoder and focus learning on the encoder (rewarded by downstream accuracy) is elegant. Showing near-perfect decoding when given oracle summaries isolates where the difficulty lies.

**Weaknesses:**

- The contribution “we cast knowledge state modeling as an inverse problem” overclaims a bit: this framing is long-standing in cognitive diagnosis/psychometrics. Section 2.2 centers CD and deep KT but does not mention at all about classic Bayesian KT (BKT/IRT/HKT/PSIKT)s that already posit latent states generating responses.
- Figure 4 is overloaded (many colors/markers/linestyles) and almost impossible to read given that is the main result figure, making the sample efficiency story hard to parse. Please split into subplots (by backbone or family), or group KT vs. CD vs. LBM, and simplify.
- I am a bit confused by those quite different evaluations. The decoder is prompted to output “Yes/No”, and for open-source models AUC is computed from logits of those tokens, whereas closed-source models are parsed from text. This apples-oranges setup can bias AUC and calibration. Recommend a unified probability extraction (verbalizer sets or logit bias) and reporting ECE/Brier in addition to accuracy/AUC. Also, CD is evaluated with same-student 80/20 splits, while KT/LBM use unseen-student evaluation and fixed |Y|=4. This makes cross-family comparisons too tricky to understand.
- LLMs further evaluation
    - LLMs require question text, many KT/CD datasets provide only IDs. The authors acknowledge this limitation but did not test robustness to degraded text. Please add controlled paraphrase/noise/ablation tests (e.g., ID+short stem, masked tokens) to quantify sensitivity and deployment feasibility on ID-only platforms.
    - I assume LLMs' produced summary is hugely dependent on some shallow statistics/temporal correlation in the data. Please test whether LBM-identified “understands X/Y, not Z” correlates with simple signals (e.g., per-concept past success rates, transition matrix) or add human-rater studies on faithfulness/actionability. If strong correlations exist, I am not quite sure what LBMs add beyond well-tuned regressions.
    - Without explicit priors/dynamics, LLM summaries may be temporally inconsistent (e.g., “mastered X” then “forgot X” within a short window). Classical Bayesian models enforce coherence via latent dynamics. Consider at least check the consistency, to make sure the inconsistency is more coming from prompting/LLMs randomness.
- The trained-encoder experiment (GRPO) shows more gains on synthetic data, but fine-tuning can be expensive. Do you have a continual/online training strategy (e.g., periodic LoRA updates, replay buffers, cost budgets) as student data grows, and how much is the marginal gain per additional student?

**Questions:**

Could you please answer each point in the weaknesses?

---

> ### Author Response · Authors · 2025-11-28
>
> Thank you for your insightful feedback. We respond to each of your point below.
>
> ## W1: “we cast knowledge state modeling as an inverse problem” overclaims a bit; Section 2.2 does not mention classic Bayesian KT
>
> Thank you for raising these valid concerns.
>
> We have refined our contribution to “We cast knowledge state modeling as an inverse problem **over open-ended textual representations**”. This better captures the key difference between our work and traditional cognitive diagnosis and psychometrics, which approximate knowledge states via quantitative proficiency estimates across knowledge concepts.
>
> You are right to point out that Bayesian KT methods use directly interpretable latent states. However, these latent states typically capture the student’s proficiency over knowledge concepts, similarly to CD-like diagnostic reports. We updated Section 2.2 and Appendix F to discuss this more explicitly.
>
>
> ## W2: Figure 4 is overloaded
>
> Thank you for this suggestion. We have simplified Figure 4 (Figure 8 in the updated manuscript) to only include the highest scoring model for each of LBM, KT and CD methods, and added annotations highlighting different “Trajectory Length” markers and “# Students” lines. This clarifies the figure while maintaining the main result demonstrated: a strong LBM matches the accuracy of the best KT and CD models on the Synthetic dataset with much better sample efficiency.
>
>
> ## W3: Different evaluation methodology
>
> **Difference in open- vs closed-source LLM evaluation**
>
> We acknowledge the discrepancy between the answer generation process for closed- vs open-source LLMs. This is due to the logit information not being readily available from the OpenAI API without resorting to techniques like logit bias which increases generation costs. In contrast, open-source models allow for greater control over the generation output, such as accessing the logits of specific tokens. Directly extracting the logits of Yes/No tokens is a common way to obtain AUC from LLMs, and increases the reliability of the generation process. While this means that the outputs are not strictly comparable, we argue that **this difference is minimal**, as both types of models are already prompted to answer via Yes/No already: the only difference arises from cases where the model would not have answered with either tokens, which are commonly discarded or filtered out.
>
> **Difference in CD vs KT/LBMs evaluation**
>
> Regarding the evaluation for CD, we first apologize for an inaccuracy in the original Appendix: we use a train/test split of 88/12 for EEDI and XES3G5M, to ensure that the number of train/test questions matches the KT and LBM evaluation on these datasets ($|\mathcal{X}|+|\mathcal{Y}|=34$; $34*0.12=4=|\mathcal{Y}|$). We have corrected this inaccuracy in the updated manuscript.
>
> The difference in evaluation between CD vs KT/LBM model is due to the structural differences between these methods: pretrained KT models and LBMs can be ran on a previously unseen test student’s trajectory, whereas CD models require having been trained on the student and therefore can only be tested on withheld test questions.
> While this means the evaluations are not strictly comparable, we use the above train/test split to ensure a fair comparison.

---

> > ### Author Response · Authors · 2025-11-28
> >
> > ## W4: Further LLM evaluation
> >
> > **a. Question text ablation**
> >
> > We acknowledge that LBMs require question text, which limits their applicability to ID-only platforms. In such settings, where LBMs would be restricted to insights about general knowledge concepts, traditional KT and CD models are better suited. However, most real-world educational platforms include question text, making LBMs widely applicable in practice.
> >
> > **b. Summary correlates with simple signals like per-construct accuracy**
> >
> > We agree that LBMs likely leverage signals from simple correlations, such as overall success rates on each construct. However, KT and CD models rely on these same correlations to predict knowledge states accurately. Moreover, LBMs can capture non-trivial response patterns such as misconceptions that go beyond simple per-construct accuracy.
> >
> > Thank you for suggesting a more systematic study of faithfulness/actionability. We have added **new experiments where we systematically evaluate the quality of the produced summary across dimensions such as overall alignment with ground truth/misconception detection and false positive (relating to faithfulness) and specific/calibration (relating to actionability)**. Please refer to **Section 5** and **Appendix F** in the updated manuscript for detailed methodology and results.
> >
> > Our results reveal interesting insights, such as the fact that LLM encoders can capture more misconceptions than others but at the cost of more false positives/hallucinations (Figure 4.b). Moreover, the strongest model evaluated (GPT-5) detects student misconceptions nearly 80% of the time. In contrast, KT/CD models are unable to detect specific misconception as they are typically limited to estimates of knowledge concept mastery.
> >
> >
> > **c. No dynamic modelling**
> >
> > Thank you for this suggestion. This work considers a static knowledge state, which is a common assumption in Cognitive Diagnosis. While LBMs can be extended to dynamic knowledge state settings in several ways (we refer to our response to reviewer q3Yi's Q3 for further discussion), we leave this for future work.
> >
> > ## W5: Continual learning strategy
> >
> > You raise an important point that given the computational cost of GRPO-based encoder finetuning, ensuring that encoder model can be continuously refined over time instead of retraining from scratch is essential for deployment in real educational settings where student populations evolve over time. This is outside the scope of this current work, but represents a very interesting direction for future work.

---

### Official Review · Reviewer_mQiF · 2025-10-30

**Soundness:** 3
**Presentation:** 3
**Contribution:** 2
**Rating:** 2
**Confidence:** 4

**Summary:**

The authors suggest going through an text-based intermediate representation in natural language to provide interpretable knowledge tracing. They show their method matches or underperforms existing approaches on 1 synthetic and 2 real datasets.

**Strengths:**

I felt that the proposed approach was interesting.

**Weaknesses:**

The authors keep stating that existing approaches just provide "quantitative skill mastery estimates" or "uninterpretable latent representations" but the proposed approach, which does rely on embeddings that are similar latent representations, may hallucinate, and the authors do not provide any qualitative assessment of the generated explanations.

The authors write:
> "Finally, recent LLM-based approaches have shown promise for knowledge tracing tasks (Li et al., 2024a; Wang et al., 2025), but they generally remain opaque, either treating LLMs as black boxes or relying on
model-generated explanations susceptible to hallucination."

I don't see how the proposed approach is not also treating LLMs as black boxes nor wouldn't be susceptible to hallucination.

"rigid predefined KC taxonomies" is too vague (I suspect this is generated by LLM), and repeated over the text. I assume the authors mean that the q-matrix needs to be provided, but as the authors state it themselves, some neural approaches for cognitive diagnosis can learn the q-matrix.

"unintepretable latent representations" But nothing prevents the authors from trying to interpret a posteriori a learned vector by an existing deep learning approach for knowledge tracing.

> "The largest performance gap arises on XES3G5M. However, this dataset has an average accuracy of 85%, implying that even a constant predictor would achieve 85% accuracy."

One way to avoid this is looking at AUC, which is what is done in Table A1.
It also means that the proposed LLMs are performing worse than a constant predictor (with respect to accuracy).
Table A1 seems to indicate that models like gemma-3-27b seem to have 0.78 AUC which is among the top AUC, while their accuracy is .33 among the worst one. This should be discussed.

In the appendix:
> Compare to Cognitive Diagnosis which assumes a constant knowledge state, Knowledge Tracing method aim at estimating evolving knowledge states as students answer questions. We similarly review

This sentence seems incomplete. Also it should start with "Compared". Another typo: "[KT] methods".

Minor: in section 1358 the authors put IKT and QIKT in the same paragraph but those models are very different, and QIKT is not meant to be interpretable.

**Questions:**

In the synthetic dataset, who wrote the ground truth knowledge? Is it curated by a human or yet another LLM?

---

> ### Author Response · Authors · 2025-11-28
>
> Thank you for your detailed and constructive comments. We address each of your points below.
>
> ## W1a & W2: LBMs still rely on embeddings and may hallucinate
>
> Our method is based on LLMs which indeed use embeddings to represent text internally. However, the language bottleneck in our architecture enforces that **all the information used by the decoder to predict future answers must be passed on via natural language**. While this does not prevent the encoder from hallucinating, it means that the accuracy of the overall model is directly tied to the ability of the encoder to faithfully represent the student’s knowledge state in natural language. In contrast, the mentioned works using LLMs for KT (Li et al., 2024a; Wang et al., 2025) directly predict future answers without providing any interpretable intermediate representations. We have updated the manuscript (e.g l.262) to better reflect this point.
>
> ## W1b: No qualitative assessment of the generated explanations
>
> We agree that a systematic qualitative assessment of the generated explanations is crucial to our evaluation. To this end, **we have added new experiments that systematically evaluate the quality of produced summaries on the Synthetic dataset across various axes using LLM-as-a-judge with GPT-5**. Detailed evaluation methodology, results and illustrative examples are provided **Section 5** and **Appendix C** in the updated manuscript.
>
> Notably, results illustrate how different models are differently susceptible to hallucinations (ie misconception false positive), with Gemma3 models detecting more misconceptions than GPT-4o/4o-mini but at the cost of more misconceptions (**Section 5.1**). Moreover, training the encoder improves the general accuracy of the summaries, as well as their specificity and calibration (**Section 5.2**).
>
>
> ## W3: "rigid predefined KC taxonomies" is too vague
>
> Thank you for pointing out this lack of clarity. By this term we mean not only that a Q-matrix must be provided or learned, but more generally that the diagnostic report is directly tied to the set of knowledge constructs considered. As a result, **even if the Q-matrix is learned from data, the diagnostic report remains tied to a fixed set of knowledge constructs**. We have updated the language throughout the manuscript to clarify this point.
>
> ## W4: Learned latent vectors can be interpreted a posteriori
>
> This is a valid point, but it transfers the burden of interpretability to post-hoc interpretability analysis. Simple post-hoc analyses—such as gathering predictions on a wide set of questions and reporting predicted correctness rates across constructs— suffers from the same limitations as CD-like diagnostic reports: being constrained to mastery estimates over a fixed set of knowledge concepts. In contrast, LBMs offer interpretability-by-design by enforcing that all predictive information from the encoder must be captured in natural language.
>
>
> ## W5: Class imbalance in XES3G5M; AUC/Accuracy discrepancy in Table A1
>
> This is indeed why we report AUC in Table A1. We chose to focus on showing accuracy results in Section 5 since we cannot access logits and therefore AUC for the closed-source GPT models.
>
> Thank you for pointing out the surprisingly low accuracy but strong AUC obtained by some of the models. This discrepancy arises because here **LBMs are evaluated zero-shot and therefore cannot leverage dataset-level statistics such as class proportions**. As a result their predictions might be skewed towards the minority class, while the EEDI and XES3G5M datasets are imbalanced towards the positive class (e.g. 85% for XES3G5M). This explains why some models achieve worse accuracy than a constant predictor while still maintaining strong AUC scores--the AUC being high shows that the model still captures meaningful predictive signals. We have clarified this point in Appendix A.2.
>
> ## W6 & W7: Incomplete sentence/typo/related works organization
>
> Thank you for catching these oversights. We have corrected them in the updated manuscript. We now separately discuss IKT as a Bayesian method for KT, and QIKT in the "IRT-based Knowledge Tracing" paragraph.
>
>
> ## Q1: In the synthetic dataset, who wrote the ground truth knowledge?
>
> The Synthetic dataset is generated programmatically: for each student, we randomly sample knowledge concept masteries and misconceptions from a pre-defined set. We then generate each student’s answers to questions based to (1) whether or not they mastered the question’s construct, and (2) whether or not any of their misconceptions apply. Details about the generation process of our Synthetic dataset are provided Appendix B.3.1.
>
>
> Thank you again for your comments. We believe these revisions have improved the evaluation of our method and clarified our contribution relative to existing works, and hope our response addresses your concerns.

---

### Official Review · Reviewer_n3oN · 2025-10-31

**Soundness:** 2
**Presentation:** 2
**Contribution:** 2
**Rating:** 4
**Confidence:** 4

**Summary:**

This paper presents a project on creating a model -- a language bottleneck model, for analyzing students' skills and misconceptions through a combination of two large language models. The first model serves as the encoder, encoding the students' past learning history into natural language explanations, and then uses another LLM to decode it into the original submissions as well as predictions on the next submissions. They performed predictions on a synthetic dataset, as well as two public datasets, and showed that overall the results are better in most base models.

**Strengths:**

+ Overall this paper is relatively easy to follow and read. The motivation is easy to understand.
+ The design of the model is straightforward. Under the context provided by the authors, the design makes sense.

**Weaknesses:**

- One major issue of the paper is about evaluation.
 -> 1) While the design of the model is centered around interpretability (multiple places are showing this, including introduction, discussions, and the design considerations, etc.), there is no systematic evaluation of this perspective. While the case studies give a peek at the performance, and it is looking good, it still lacks formal evaluations. In some cases, it might be a case that some natural language summarization could be incorrect or not interpretable, but still decoded correctly. A more careful look is needed.
 -> 2) The result of accuracy, if interpreted correctly, seems to be similar to CDM, even with large language models. Although it saves the total number of seen questions, it is still not a major improvement motivated in this work.
Overall, the work is interesting, and the results may use better presentations to be more relevant to the motivations of the work.
- There are also some minor issues with the narratives of the work, listed in the questions below.

**Questions:**

- Line 12: The goal of KT, though, is still to estimate students' skills.
- Line 15: This only applies to DKT models. For BKT models, they have clear interpretability.
- Line 41: Again, the one you cited from Corbett is BKT, and it does not have vector representations for knowledge -- it's just a set of statuses representing whether students know certain knowledge or not. It is quite interpretable.
- Line 50: The following statement should be fine even outside of the CD domain.
- Line 170: This is not convincing -- since the decoder part can produce good results already, then why wouldn't we make it better?
- Line 177: At this point, readers start to wonder what exactly the knowledge state will look like in natural language. For some tasks, like open-ended problems, it is just hard to reconstruct the exact same answers, no matter how good the LLM is.
- Line 306: Preprocessing of datasets should not be in Appendix as it is necessary for replication. It is an integral part for a research paper to be validated.
- Line 469: So the quality of the summary should be systematically evaluated.

---

> ### Author Response · Authors · 2025-11-28
>
> Thank you for your insightful and constructive feedback. We appreciate your emphasis on systematic interpretability evaluation, which is indeed central to our work's contribution. We have substantially revised the manuscript to address your suggestions, with a major restructuring of Section 5 to place qualitative evaluation of LBM summaries in the foreground. We address your specific comments in detail below.
>
> ## Lack of systematic evaluation of interpretability
> We agree that systematically evaluating LBM summaries and providing insights about the interpretability of our approach is crucial. Moreover, as you rightly note, the main motivation for this work is not pure accuracy-based improvement over baselines, but rather the qualitative difference in knowledge state estimates produced by LBMs compared to KT/CD models. Therefore **we have significantly updated Section 5 with new experiments, as well as moving qualitative evaluation of LBMs to the foreground of section to reflect this focus**.
>
> We now systematically evaluate the summaries produced by different LLM encoders for Synthetic students using LLM-as-a-judge. Specifically, we provide GPT-5 with a produced summary alongside the ground truth knowledge state, and prompt it to evaluate the summary’s general alignment with ground truth, construct mastery accuracy, misconception detection and false positives, specificity and confidence calibration. See **Appendix C** for evaluation details, prompts and examples.
>
> Results on different LLM encoders (**Section 5.1**) show that more capable models generally produce summaries that are more aligned with the ground truth and capture concept mastery more accurately, as well as being more specific and better calibrated. Interestingly Gemma3 models can better detect misconceptions than GPT-4o/4o-mini but at the cost of more hallucinations (ie mentioning misconceptions not present in the ground truth), while GPT-5 outperforms other models across the board.
> Additionally, results on the trained encoder (**Section 5.2**) show that the alignment with ground truth, construct mastery accuracy, specificity and calibration all gradually improve over the course of training.
>
> ## Minor issues with the narrative
> Thank you for these precise suggestions, which have significantly improved the narrative and clarity of our work. We have updated the manuscript to address each point:
> - l.12: revised to "Cognitive Diagnosis (CD) models estimate student proficiency at a fixed point in time, while Knowledge Tracing (KT) methods model evolving knowledge states to predict future performance."
> - l.15 & l.41: we now distinguish between deep learning-based KT and more interpretable KT methods like BKT.
> - l.50: generalised beyond the CD domain
> - l.170: you are right that training the decoder to further improve its performance is still valuable. We updated the text to make clear that this is a decision for the scope of the current work.
> - l.177: we now refer to Figure A1 in the appendix for example knowledge state/summaries, and specify that we consider closed-form questions.
> - l.306: moved dataset preprocessing details from the Appendix to the main text.
> - l.469: we hope the new systematic evaluation of summaries above addresses this concern.
>
> These revisions have significantly strengthened the manuscript's focus on interpretability, and we hope they fully address your concerns. Thank you again for your valuable feedback.

---

### Official Review · Reviewer_q3Yi · 2025-11-01

**Soundness:** 2
**Presentation:** 2
**Contribution:** 2
**Rating:** 2
**Confidence:** 5

**Summary:**

This paper proposes Language Bottleneck Models (LBMs) for representing student knowledge states in natural language. The encoder LLM produces concise textual summaries of a student’s knowledge, and a decoder LLM reconstructs past responses and predicts future performance solely from that text. This transforms the traditionally quantitative representations used in cognitive diagnosis and knowledge tracing into interpretable textual summaries. The authors evaluate LBMs on synthetic, Eedi, and XES3G5M datasets, showing that zero-shot LBMs achieve performance comparable to state-of-the-art KT/CD models while offering interpretability and qualitative insight into misconceptions. The paper further explores reinforcement learning to refine summaries and demonstrates steerability through prompt or reward shaping

**Strengths:**

1. The encoder–decoder LLM design is interesting, especially that the decoder reconstructs past responses and predicts future performance solely from the textual bottleneck.
2. The work demonstrates a strong theoretical framing that connects cognitive diagnosis, knowledge tracing, and language bottlenecks in a coherent way, supported by extensive experiments across diverse datasets.

**Weaknesses:**

1. The knowledge state representation defined by coarse textual categories such as Mastered, Fails on, and Misconceptions (Figure 5) may lose important intermediate information—for instance, differences between mastery levels of 0.6 and 0.7. Moreover, extracting precise concept-level interpretations from free-form text can be ambiguous due to synonymy and linguistic variability.
2. The compared CD and KT baselines do not include recent LLM-based variants[1,2,3], which may lead to an incomplete assessment of the proposed method’s relative performance and limit the fairness of the comparison.
3. There has limited analysis on scalability and cost of the two-stage LLM setup in real deployments.
4. Some sections (e.g., 5.4) could include more statistical rigor on variance and significance.
5. Missing user or teacher evaluation of the interpretability claims (qualitative human study).
6. The code is not released, which may lead to difficulties in reproducing the paper

```
[1] Wang Z, Zhou J, Chen Q, et al. LLM-KT: Aligning Large Language Models with Knowledge Tracing using a Plug-and-Play Instruction[J]. arXiv preprint arXiv:2502.02945, 2025.
[2] Dong Z, Chen J, Wu F. Knowledge is power: Harnessing large language models for enhanced cognitive diagnosis[C]//Proceedings of the AAAI Conference on Artificial Intelligence. 2025, 39(1): 164-172.
[3] Li H, Yu J, Ouyang Y, et al. Explainable few-shot knowledge tracing[J]. Frontiers of Digital Education, 2025, 2(4): 34.
```

**Questions:**

1. How does the method handle long student histories given LLM context limits?
2. Would joint training of encoder and decoder yield better interpretability or stability?
3. Could LBMs extend to evolving knowledge states (non-static) for longitudinal modeling?
4. What are the compute and cost implications compared to KT baselines for large-scale deployments?

---

> ### Author Response · Authors · 2025-11-28
>
> Thank you for your constructive feedback. We appreciate your positive assessment of our method and framing. We have significantly updated the manuscript following your suggestions, including **new systematic evaluations of the produced summaries, improved statistical rigor, and an explicit analysis of the computational cost of our method and baselines**.
>
> We address your comments in detail below.
>
> ## Response to W1
> The “True Knowledge State” in our case-study (previously Figure 5, now Figure 3 in the updated manuscript) corresponds to the ground truth knowledge of a synthetically generated student. We provide full details of our Synthetic dataset construction in Appendix B.3.1. While this synthetic dataset is indeed a simplification of real-life educational datasets, it serves to represent nuanced traits like misconceptions that most KT/CD models cannot capture. **We emphasize that the free-form textual knowledge states generated by our model are not limited to these coarse categories**.
>
> You are correct that free-form text can be less precise than quantitative estimates of concept mastery as produced by KT and CD models. However, it provides the flexibility to capture nuanced behaviors like misconceptions and therefore is a valuable orthogonal source of insight for teachers.
>
>
> ## Response to W2
> Thank you for suggesting these references. We already discuss [1] and [3] in Section 4 and Appendix F.2.3 (previously D.2.3): while they show that LLMs can be used as powerful knowledge tracing models, both methods use LLMs as direct predictors of future student answers, without any interpretable intermediate representation. Therefore, they suffer from the same limitations as deep KT models in terms of interpretability.
>
> While [2] leverages the general knowledge of LLM to refine a cognitive diagnosis model, it operates within the traditional framework of cognitive diagnosis: the method produces a diagnostic report of mastery levels across knowledge concepts. In contrast, our method allows for insights on more nuanced knowledge traits like misconceptions by representing knowledge states in free-form text. We have updated Appendix F.2.3 to include a discussion this work.
>
>
> ## Response to W3 & Q4
> Thank you for this suggestion. We have added an analysis comparing the computational costs of LBMs against KT, CD, and Direct baselines over all our datasets in Appendix E.
>
> ## Response to W4
> We have updated Section 5.4 (now Section 5.5) to use Welch’s t-test with a one-sided hypothesis to identify runs for which the LBM accuracy is no more than 2% below the direct accuracy with the same LLM backbone.
>
> ## Response to W5
> While a formal qualitative human study is outside the scope of this work, we agree that a more systematic evaluation of the produced summary is critical to validate our method.
>
> To this end, we have added **new experiments evaluating the quality of the LBM summaries against ground truth** (available in our Synthetic dataset), using LLM-as-a-judge with GPT-5. LLM-as-a-judge has been shown to be a reliable proxy for human evaluation in the NLP literature [4], and provides several advantages here: it enables systematic evaluation across multiple dimensions at scale, and our synthetic dataset allows us to compare against oracle knowledge states.
>
> Specifically, we evaluate the general alignment with ground truth, construct mastery accuracy, misconception detection and false positives, as well as the specificity and confidence calibration of the summary produced by different encoder LLMs. This new evaluation is detailed in **Section 5** and **Appendix C** of the updated manuscript.
>
> Our new results from **Section 5.1** show that model families and capabilities performance vary across these axes, with GPT-5 emerging as a clear winner across all dimensions. Moreover, **Section 5.2** shows that our GRPO-based encoder training procedure leads to improved alignment and construct accuracy, as well as higher specificity and better calibration.
>
> ## Response to W6
> We will be releasing the code for our method and experiments upon acceptance.

---

> > ### Author Response · Authors · 2025-11-28
> >
> > ## Response to Q1
> > You rightfully point out that the current method is limited by the encoder’s context limits.
> > However, the encoder-decoder architecture of LBMs allows to easily extend the effective context size of the model: long student histories can simply be split into chunks of questions and passed to the encoder while the textual summary is iteratively refined. This also opens up possibilities for active learning, where the model would select the most informative questions to refine the textual summary. We leave these interesting directions for future work.
> >
> >
> > ## Response to Q2
> > In this work we choose to freeze the decoder model and focus on the encoder since the task of the encoder is much harder than that of the decoder (as illustrated by our motivating observations in Section 3.2). Nevertheless, the decoder could be trained jointly with the encoder to improve the accuracy of the decoding and therefore provide more accurate reward signals to the encoder. Jointly training the encoder and decoder would consist in alternately taking GRPO steps to update the encoder model with a fixed decoder and then training the decoder on summaries produced by the fixed encoder. This is an exciting next step for LBMs which we leave for future work.
> >
> >
> > ## Response to Q3
> > This work focuses on static knowledge states -- as commonly done for CD models -- but LBMs can be extended to non-static settings in several ways. A simple way would be to provide the encoder with time information about when each question was answered, and let the encoder LLM reason about any evolution in the student's knowledge state (eg successfully learning a concept then forgetting it later). Alternatively, one could split the student history by chunks of questions answered at similar times and encode each one iteratively while providing the encoder with past summaries and time information for each chunk, producing a longitudinal trajectory of textual knowledge states. We leave these developments for future works.
> >
> >  ---
> >
> > [4] Li, Dawei, et al. "From generation to judgment: Opportunities and challenges of llm-as-a-judge." Proceedings of the 2025 Conference on Empirical Methods in Natural Language Processing. 2025.

---

### Author Response · Authors · 2025-12-02
**Summary of Revisions**

We thank all the reviewers for their constructive feedback. We understand that the primary concern across reviews (*Reviewers q3Yi, n3oN, mQiF, aM4W*) was the need for a **systematic evaluation of the interpretability and quality of knowledge state summaries produced by LBMs**, beyond the illustrative case-study in our original submission.

To address this, we have introduced a **new evaluation framework** using LLM-as-a-judge (GPT-5) to systematically assess summary quality on our Synthetic dataset across multiple dimensions: general alignment with ground truth, construct mastery accuracy, misconception detection and false positives, specificity, and confidence calibration. This evaluation covers both zero-shot encoders (**Section 5.1**) and GRPO-trained encoders (**Section 5.2**). Our results reveal some interesting trade-offs across encoder LLMs: for instance, Gemma3 models detect more misconceptions than GPT-4o but produce more false positives, while GPT-5 achieves strong performance across all dimensions, detecting student misconceptions nearly 80% of the time. Our GRPO-based training further improves general alignment with ground truth, construct accuracy, specificity, and calibration. **Critically, this ability to explicitly identify specific misconceptions is fundamentally lacking from the quantitative proficiency estimates of traditional KT/CD models.**

Additional revisions in response to the reviewers’ feedback include:
- **Refined contribution statement and related work** (*n3oN, mQiF, aM4W*): Clarified contribution as casting "knowledge state modeling as an inverse problem *over open-ended textual representations*" and expanded discussion of Bayesian and LLM-based KT methods (Section 2.2, Section 4, Appendix F),
- **Enhanced statistical rigor** (*q3Yi*): Added Welch's t-test analysis in Section 5.5,
- **Computational cost analysis** (*q3Yi*): New Appendix E comparing the computational cost of LBMs against baselines,
- **Improved clarity and figures** (*n3oN, aM4W*): Moved dataset preprocessing to main text, simplified Figure 8.

These revisions have significantly improved our manuscript, and we are grateful to the reviewers for their engagement in this process.

---

### Meta-Review · Area_Chair_VSTj · 2026-01-02

**Summary:**

1. Limited evaluation and missing human-experiments-based evaluation (q3Yi, n3oN, mQiF, aM4W)
All reviewers have raised concerns about the unconvincing evaluation, conducted in an overly simplified setting. The paper states a contribution that uses language models to facilitate education. However, there is no human-subjects-based evaluation to validate the effectiveness of the proposed method in real educational settings, despite the authors' determination that such an evaluation is out of scope. This is the primary limitation of the paper, as it leaves considerable uncertainty about the impact. This is the major reason for me to recommend a rejection of this paper.

2. The presentation could be further improved (aM4W, n3oN).

3. Other minor issues such as statistical rigor (q3Yi), unfair narratives or discussion regarding related work (q3Yi, n3oN, mQiF, aM4W).

**Reviewer Concerns:**

I don't believe Concerns 1 and 2 above have been adequately addressed. Concern 3 is somehow addressed during rebuttal.

**Reviewer Scores:**

Based on my educated guess, the authors' responses will not change any of the reviewers' opinions to cross the acceptance/rejection boundary, although there is a considerable chance for Reviewer n3oN to raise the rating to 4, which will still not change my recommended decision.

---

### Decision · Program_Chairs · 2026-01-26

Reject